# LaRA-Fusion: Latent-Robust Adaptation via Dual-Loop Constraints for Infrared and Visible Image Fusion

Yaru Su [* 1 2]  Chaowei Huang [* 1 2]  Huangbiao Xu [1 2]  Xiao Ke [1 2]

## Abstract

Infrared and visible image fusion (IVIF) aims to synergize complementary thermal radiation and textural details for comprehensive scene perception. However, existing unsupervised paradigms often overlook the intrinsic topological consistency shared across modalities. Lacking explicit geometric regularization, encoders frequently succumb to degenerate numerical shortcuts, capturing superficial high-frequency noise rather than domain-invariant semantic structures to satisfy reconstruction objectives. To address this, we propose LaRA-Fusion, a framework achieving Latent-Robust Adaptation via Dual-Loop Manifold Constraints. We construct a strictly constrained latent space where an inner loop enforces geometric reversibility, while an outer loop aligns the generated representations with the target data distribution. This mechanism effectively mitigates latent space collapse, compelling the model to extract topologically aligned features that remain robust against modality-specific variations. Extensive experiments demonstrate that LaRA-Fusion outperforms state-of-the-art methods with superior robustness and interpretability.

## 1. Introduction

In the era of multimodal learning (Xu et al., 2025b; 2026), visual perception systems are inherently constrained by the physical limitations of single-modality sensors (Ma et al., 2019a; Zhang et al., 2021). While visible (VIS) light sensors capture rich textural and color details, their strict reliance

*Equal contribution [1]Fujian Provincial Key Laboratory of Networking Computing and Intelligent Information Processing, College of Computer and Data Science, Fuzhou University, Fuzhou 350108, China [2]Engineering Research Center of Big Data Intelligence, Ministry of Education, Fuzhou 350108, China. Correspondence to: Xiao Ke <kex@fzu.edu.cn>.

*Proceedings of the $43^{rd}$ International Conference on Machine Learning*, Seoul, South Korea. PMLR 306, 2026. Copyright 2026 by the author(s).

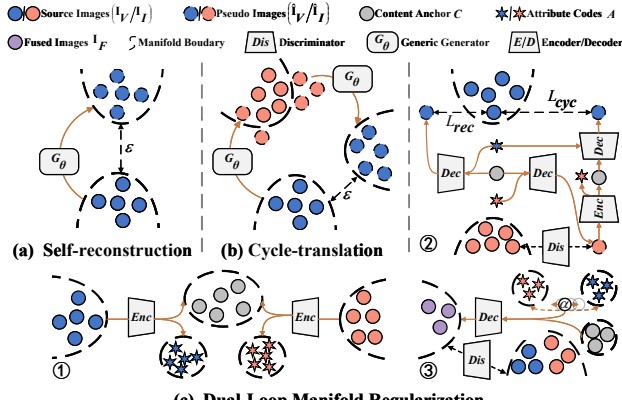

Figure 1. **Latent Robustness via Manifold Regularization. (a) Self-Reconstruction.** Generators optimized solely by pixel-wise losses often degenerate into identity mappings, ignoring semantic structure. **(b) Cycle-Translation.** Rigid cycle-consistency drives models to embed source details as imperceptible high-frequency noise in pseudo-images, causing off-manifold drift. **(c) LaRA-Fusion. (1) Disentanglement:** Inputs decompose into shared content (*C*) and specific attributes (*A*). **(2) Dual-Loop Constraint:** An inner loop enforces geometric reversibility, while an adversarial outer loop aligns translations to the target distributions, jointly mitigating degenerate learning paths. **(3) Stochastic Manifold Traversal:** Structural stability is verified via random $\alpha$-interpolation across the manifold to promote robust fusion.

on reflected light renders them susceptible to adverse illumination or smoke (Ma et al., 2021). Conversely, infrared (IR) sensors offer robust all-weather monitoring by detecting thermal radiation, yet they inherently suffer from low resolution and a lack of geometric texture details (Yang et al., 2021). Consequently, integrating the detail-rich VIS modality with the thermally robust IR modality is imperative to overcome these intrinsic spectral trade-offs.

To surmount these spectral limitations, infrared and visible image fusion (IVIF) (Sun et al., 2020; Luo & Luo, 2023; Lu & Shi, 2024) serves as a fundamental bridge. By explicitly coupling thermal saliency with visible textural details, IVIF aims to construct a composite representation that transcends the capabilities of individual sensors. The resulting information-rich fused imagery is indispensable for advanced tasks like military surveillance (Bhavana et al., 2022). Crucially, in safety-critical scenarios like autonomous driv-

ing (Yuan et al., 2023; Li et al., 2023b), high-quality fusion goes beyond visual fidelity to serve as a prerequisite for accurate perception. By enhancing object detection (Wang et al., 2023) and semantic segmentation (Tang et al., 2022a), IVIF facilitates robust decisions in dynamic scenes.

Despite the significant advancements in IVIF, the field faces a fundamental dilemma: the absence of ideal ground truth necessitates reliance on unsupervised objectives. While early approaches depended on intricate hand-crafted losses or self-reconstruction autoencoders (Li & Wu, 2018; Zhao et al., 2020; Li et al., 2021), recent methodologies (Xu et al., 2022; Wang et al., 2022) have advanced towards translation-based frameworks. By explicitly exploiting image translation loops or generative paradigms to synthesize pseudo-modalities, these works successfully bridge the domain gap, thereby mitigating spatial misalignments and enhancing feature consistency. However, a subtle yet critical vulnerability persists within these translation-driven frameworks: the susceptibility to degenerate numerical shortcuts. Under the strict pressure to minimize translation or reconstruction errors without explicit manifold regularization, encoders tend to embed source-specific high-frequency details as imperceptible noise within the intermediate representations. Although this pathological optimization behavior satisfies numerical objectives and aids in superficial alignment, it compromises the topological integrity of the latent space. Consequently, the learned representations often lack the semantic robustness to withstand attribute variations, rendering the fusion process fragile when the rigid distribution assumptions are disrupted by complex environmental shifts.

To address this limitation and preserve the topological integrity of the latent space, we propose **LaRA-Fusion**, a novel framework governed by dual-loop manifold constraints. As illustrated in Figure 1(c), our approach departs from monolithic feature extraction by explicitly projecting inputs into decoupled subspaces: a shared, modality-agnostic content anchor ($C$) and distinct attribute coordinates ($A$). Unlike previous methods that prioritize pixel-level reversibility, we construct a dual-loop framework to constrain these representations. The inner loop enforces geometric invertibility through self-reconstruction, relying on attribute-modulated synthesis to suppress decoding artifacts. Simultaneously, the outer loop serves as a statistical regularizer, deploying adversarial discriminators on intermediate translation states to optimize the generator for rendering semantically valid textures rather than encoding imperceptible high-frequency artifacts. This adversarial pressure guides the translated features to align with the target distribution, effectively pruning the pathways previously exploited for degenerate minimization. Furthermore, to enhance the robustness of the content anchor against environmental shifts, we incorporate a stochastic manifold traversal mechanism. By traversing the attribute manifold via stochastic interpola-

tion (visualized as the $\alpha$-sampling in Figure 1), we simulate diverse imaging conditions during training. This compels the model to decouple the immutable geometric structure from transient attribute variations, promoting a structurally aligned fusion grounded in a robust, topologically consistent latent space. Our contributions are:

- We propose **LaRA-Fusion**, a framework employing a dual-loop mechanism that couples geometric invertibility with adversarial alignment. This design effectively mitigates degenerate numerical shortcuts, compelling the encoder to learn **robust, modality-invariant structural representations**.

- We introduce a specialized architecture featuring a **Manifold Rectification Block (MRB)** to alleviate latent topological inconsistencies via residual aggregation, and a **Modulated Synthesis Block (MSB)** that utilizes attribute-injected weight modulation to promote high-fidelity reconstruction.

- We devise a hybrid training strategy powered by **Hierarchical Attribute Arbitration (HAA)**. It dynamically balances multi-scale feature integration and incorporates randomized attribute sampling, regulated by adaptive regularization to maintain structurally consistent and continuous attribute interpolations.

## 2. Related Works

### 2.1. Deep Learning-Based IVIF

The landscape of IVIF has been profoundly reshaped by deep learning, evolving from early autoencoder-based paradigms to sophisticated generative and transformer-based architectures. Early frameworks, such as DenseFuse (Li & Wu, 2018) and RFN-Nest (Li et al., 2021), typically adopted a two-stage strategy, decoupling feature extraction via CNNs from hand-crafted fusion rules. While efficient, this separation limited the capability to learn complex, non-linear cross-modal interactions. Subsequently, GANs were introduced to enforce distributional constraints. Methods like FusionGAN (Ma et al., 2019b) modeled fusion as an adversarial game, compelling results to retain specific spectral characteristics. Recognizing the restricted receptive fields of CNNs, recent methodologies have integrated Transformers (Ma et al., 2022; Tang et al., 2023) to capture long-range dependencies (Xu et al., 2024a; 2025a;c). Furthermore, a growing trend has shifted towards task-driven fusion (Liu et al., 2022; Wu et al., 2023), incorporating high-level semantic losses to benefit downstream perception. Despite these advancements, most existing IVIF approaches prioritize pixel-wise reconstruction, often neglecting the topological consistency of the latent space. Lacking explicit manifold regularization, deep encoders are prone to degenerate numerical shortcuts, capturing imperceptible high-frequency artifacts rather

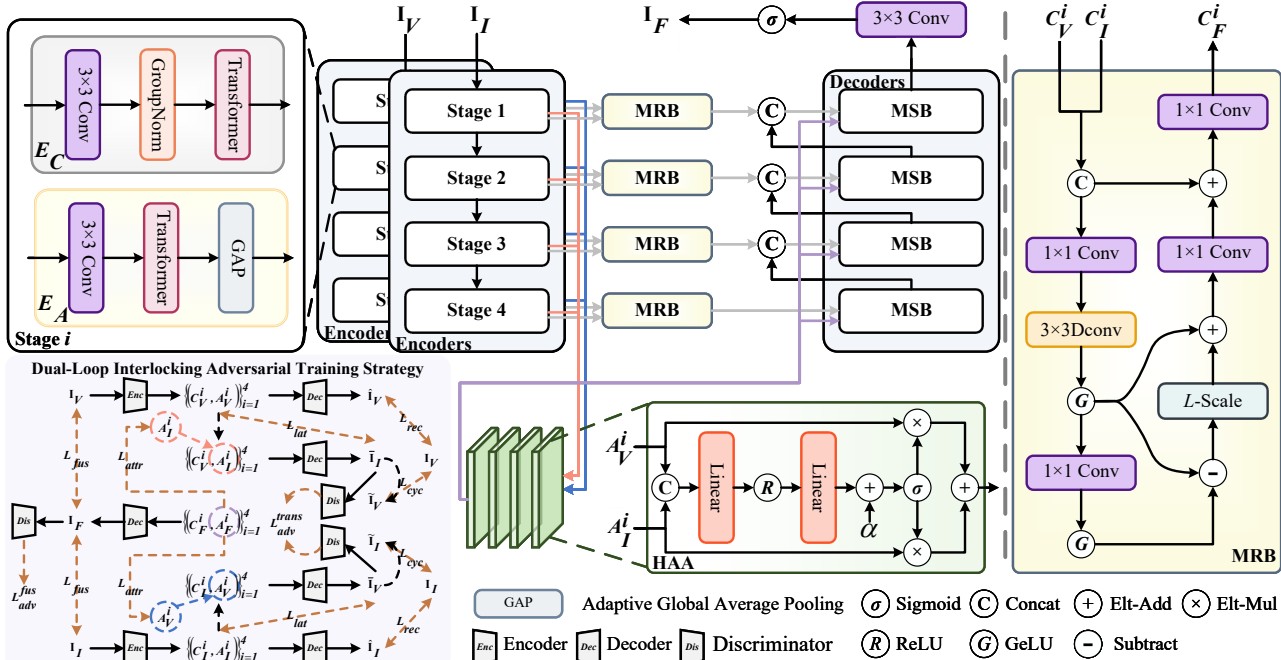

*Figure 2.* **The overall framework of LaRA-Fusion.** The architecture comprises a dual-stream multi-scale encoder and a decoder constructed from cascaded **Modulated Synthesis Blocks (MSBs)**. Content features are aggregated via **Manifold Rectification Blocks (MRBs)** across skip connections to alleviate topological misalignment. The **Hierarchical Attribute Arbitration (HAA)** module dynamically balances multi-modal attribute vectors modulated by the stochastic interpolation factor $\alpha$, subsequently driving the weight modulation within the MSBs. The **Training Strategy** panel (bottom-left) illustrates the dual-loop interlocking mechanism and the topological constraints employed to enforce latent robustness.

than domain-invariant semantics to minimize numerical errors. This pathological optimization often renders the fusion framework fragile against environmental shifts. In contrast, LaRA-Fusion introduces dual-loop manifold constraints to mitigate these degenerate pathways, promoting robust topological consistency.

## 2.2. Disentanglement and Manifold Regularization

**Disentangled Representation.** Disentanglement aims to separate content ($C$) from attributes ($A$) (Huang et al., 2018), enabling controllable IVIF by isolating thermal features from visible illumination (Xu et al., 2021). This separation facilitates the effective integration of thermal saliency and visible textures. Standard approaches rely on cycle-consistency (Zhu et al., 2017) to impose reversibility constraints in the absence of ground truth.

**Latent Manifold Regularization.** Sole reliance on reconstruction loss often leads to degenerate numerical shortcuts (Ilyas et al., 2019), where models encode source details as imperceptible noise to satisfy numerical objectives without true geometric alignment. Although topological consistency has been explored to preserve latent structure in manifold learning (Moor et al., 2020), explicit manifold regularization to prevent such latent collapse remains underexplored.

## 3. Method

### 3.1. Methodological Overview

As illustrated in Figure 2, we propose **LaRA-Fusion**, an unsupervised framework establishing a dual-loop latent space to couple geometric invertibility with distribution alignment, thereby mitigating degenerate numerical shortcuts. The framework utilizes a multi-scale architecture where single-channel source images $I \in \{I_{\mathcal{I}}, I_{\mathcal{V}}\} \subset \mathbb{R}^{H \times W \times 1}$ are decomposed into decoupled representations: a shared **content anchor** ($C$) and distinct **attribute coordinates** ($A$). To ensure topological integrity, we devise a rigorous interlocking workflow. The inner loop enforces geometric reversibility through cyclic mapping (i.e., $I_{\mathcal{V}} \rightarrow \bar{I}_{\mathcal{I}} \rightarrow \tilde{I}_{\mathcal{V}}$), utilizing the Modulated Synthesis Block (**MSB**) to enforce strict latent space consistency. Complementarily, the outer loop guides the intermediate translations ($\bar{I}$) toward the target data distribution via adversarial alignment, effectively mitigating the encoder's reliance on out-of-distribution artifacts. Finally, the fusion process incorporates the Manifold Rectification Block (**MRB**) for residual feature aggregation, governed by Hierarchical Attribute Arbitration (**HAA**). This mechanism dynamically balances feature integration with stochastic attribute interpolation, enhancing structural stability against environmental shifts.

## 3.2. Multi-Scale Disentangled Encoding

Given the source images $I_\mathcal{V}, I_\mathcal{I}$, we employ a hierarchical dual-stream framework to project them into a disentangled latent space. Leveraging a hybrid CNN-Transformer architecture at each scale $i$, the encoding process branches into a weight-shared **Content Encoder ($E_C$)** and modality-specific **Attribute Encoders ($E_A$)**. Formally, for a modality $\mathcal{M} \in \{\mathcal{V}, \mathcal{I}\}$, the feature extraction is defined as:

$$C^i_\mathcal{M} = E^i_C(I_\mathcal{M}), \quad A^i_\mathcal{M} = E^i_A(I_\mathcal{M}), \tag{1}$$

where $C^i_\mathcal{M} \in \mathbb{R}^{C_i \times H_i \times W_i}$ denotes the spatial feature map preserving geometric structures, while $A^i_\mathcal{M} \in \mathbb{R}^{C_i}$ represents the global attribute vector summarizing the statistics.

## 3.3. Manifold Rectification Block

To mitigate topological misalignments between modalities, we employ the **Manifold Rectification Block (MRB)**. Unlike direct concatenation which may yield a discontinuous latent surface, MRB refines the joint feature space through a detail-preserving residual mechanism.

Let $C^i_\mathcal{V}, C^i_\mathcal{I} \in \mathbb{R}^{C_i \times H_i \times W_i}$ denote the disentangled content features. We first construct a joint hypothesis $C^i_{cat} = [C^i_\mathcal{V}, C^i_\mathcal{I}]$. To capture inter-modal dependencies, $C^i_{cat}$ is projected into a high-dimensional manifold to facilitate channel interaction, formulated as:

$$H^i_{proj} = \mathcal{G}\left(\mathrm{Conv}_{dw}\left(\mathrm{Conv}_{in}(C^i_{cat})\right)\right), \tag{2}$$

where $\mathrm{Conv}_{in}$ is a $1 \times 1$ convolution for feature expansion, $\mathrm{Conv}_{dw}$ denotes depth-wise convolution for spatial aggregation, and $\mathcal{G}(\cdot)$ represents the GELU activation.

Within this manifold, we formulate a learnable mechanism for frequency separation to selectively enhance high-frequency textures. Specifically, we estimate a latent low-frequency component $B^i$ using a channel compressor $\phi(\cdot)$, and subsequently amplify the high-frequency residuals:

$$\begin{aligned} B^i &= \mathcal{G}(\phi(H^i_{proj})), \\ H^i_{enh} &= H^i_{proj} + \mathcal{S}^i \odot \left(H^i_{proj} - B^i\right). \end{aligned} \tag{3}$$

Here, $\mathcal{S}^i$ denotes the learnable scale factor (corresponding to L-Scale in Figure 2) that adaptively weights the extracted details ($H^i_{proj} - B^i$) to fill textural deficiencies. Finally, the rectified features are fused with the initial hypothesis via a residual connection to preserve geometric integrity:

$$C^i_\mathcal{F} = \mathrm{Conv}_{fuse}\left(C^i_{cat} + \mathrm{Conv}_{out}(H^i_{enh})\right), \tag{4}$$

where $\mathrm{Conv}_{out}$ projects the enhanced features back to the original dimension. Crucially, this residual design establishes a semantic-preserving gradient pathway, allowing $\mathrm{Conv}_{fuse}$ to subsequently compress the aggregated features into a unified code $C^i_\mathcal{F}$. Consequently, the resulting fused manifold remains topologically compact, minimizing modality-specific discontinuities.

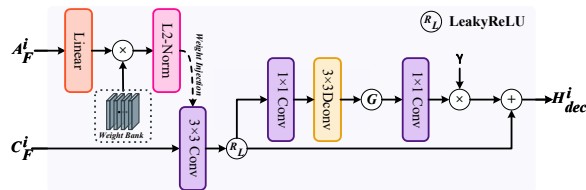

*Figure 3.* **Illustration of the Modulated Synthesis Block (MSB).** By projecting the fused attribute $A^i_\mathcal{F}$ to control a dynamic **weight bank**, the module modulates the convolution kernels to transform the fused content $C^i_\mathcal{F}$ into the decoded feature $H^i_{dec}$.

## 3.4. Hierarchical Attribute Arbitration

To synthesize a coherent attribute code robust to extreme variations, we propose the **Hierarchical Attribute Arbitration (HAA)** mechanism. Unlike static interpolation, HAA employs a latent bias modulation scheme within the unnormalized logit space, dynamically equilibrating intrinsic data priors with extrinsic stochastic cues.

Formally, given the global attribute vectors $A^i_\mathcal{V}, A^i_\mathcal{I} \in \mathbb{R}^{C_i}$ at scale $i$, we model the intrinsic fusion preference $\mathbf{u}^i_{int}$ via a lightweight bottleneck projection. This structure forces the network to distill discriminative cross-modal correlations by compressing and restoring channel interactions:

$$\mathbf{u}^i_{int} = \mathbf{W}_{up} \cdot \mathcal{R}\left(\mathbf{W}_{down} \cdot [A^i_\mathcal{V}, A^i_\mathcal{I}]\right), \tag{5}$$

where $\mathbf{W}_{down}$ and $\mathbf{W}_{up}$ denote the weights for dimensionality reduction (with ratio $r = 4$) and restoration, respectively, and $\mathcal{R}(\cdot)$ is the ReLU activation.

Instead of imposing hard constraints via linear interpolation, we treat the stochastic coefficient $\alpha$ as a prior bias in the unconstrained logit space. We employ the logit function to map the probability $\alpha \in (0, 1)$ into the unbounded logit domain $\mathbb{R}$. Formally:

$$\mathbf{u}_{bias} = \lambda \cdot \ln\left(\frac{\alpha}{1 - \alpha}\right), \tag{6}$$

where $\lambda$ scales the projection. The final gating weight $\boldsymbol{\mu}^i$ is obtained by shifting the intrinsic preference $\mathbf{u}^i_{int}$ with this injected bias:

$$\boldsymbol{\mu}^i = \sigma\left(\mathbf{u}^i_{int} + \mathbb{I}_{inj} \cdot \mathbf{u}_{bias}\right). \tag{7}$$

Here, $\mathbb{I}_{inj}$ denotes the injection indicator. This logit-domain injection acts as a soft constraint, compelling the decoder to generalize across diverse attribute distributions.

Finally, the unified attribute code $A^i_\mathcal{F}$ is obtained via adaptive aggregation:

$$A^i_\mathcal{F} = \boldsymbol{\mu}^i \odot A^i_\mathcal{V} + (1 - \boldsymbol{\mu}^i) \odot A^i_\mathcal{I}, \tag{8}$$

where $\odot$ denotes element-wise multiplication. This mechanism anchors $A^i_\mathcal{F}$ onto the intrinsic manifold, serving as a semantic stabilizer that preserves intrinsic textural fidelity even under the diverse states of stochastic interpolation.

## 3.5. Modulated Synthesis Block

To effectively reconstruct spatial features, we propose the **Modulated Synthesis Block (MSB)**. This module is designed to achieve deep attribute injection while preserving the structural integrity of the content features. Instead of treating attribute infusion and spatial refinement as separate stages, the MSB integrates attribute-guided kernel adaptation with a local geometric refinement network.

Specifically, we employ a weight modulation and demodulation mechanism ([Karras et al., 2020]) to strictly govern texture synthesis via decoupled attributes. The fused attribute vector $A_{\mathcal{F}}^i$ is projected to modulate the convolutional weights, which subsequently undergo L2-norm demodulation. This design effectively suppresses the characteristic droplet artifacts observed in instance normalization techniques and ensures consistent signal magnitudes. The intermediate feature $H_{mod}^i$ is then generated by convolving the input content $C_{\mathcal{F}}^i$ with these adapted weights.

To complement this global attribute injection with local spatial refinement, we implement a lightweight inverted bottleneck structure directly following the activation. The final decoded representation $H_{dec}^i$ is produced via a residual pathway regulated by a learnable scaling parameter $\gamma$:

$$
\begin{aligned}
H_{act}^i &= \mathcal{R}_{\mathcal{L}}(H_{mod}^i), \\
H_{res}^i &= \text{Conv}_{proj}\big(\mathcal{G}(\text{Conv}_{dw}(\text{Conv}_{exp}(H_{act}^i)))\big), \\
H_{dec}^i &= H_{act}^i + \gamma \cdot H_{res}^i.
\end{aligned}
\quad (9)
$$

where $\mathcal{R}_{\mathcal{L}}(\cdot)$ denotes LeakyReLU. The term $H_{res}^i$ encapsulates the inverted bottleneck block, sequentially applying pointwise expansion, depth-wise extraction, and linear projection layers. The learnable scalar $\gamma$ regulates the residual contribution, enabling the progressive integration of high-frequency details for stable feature refinement.

## 4. Loss Function

To promote the topological alignment of the latent space and the generation of high-fidelity results, we design a composite objective function. The total loss $\mathcal{L}_{total}$ is formulated as a weighted sum of three synergistic constraints: topological consistency $\mathcal{L}_{topo}$, adversarial distribution alignment $\mathcal{L}_{adv}$, and manifold regularization $\mathcal{L}_{reg}$:

$$
\mathcal{L}_{total} = \mathcal{L}_{topo} + \mathcal{L}_{adv} + \mathcal{L}_{reg}. \quad (10)
$$

**Topological Consistency Constraint ($\mathcal{L}_{topo}$).** This term is the core of our dual-loop mechanism, designed to enforce geometric invertibility and mitigate degenerate numerical shortcuts. It integrates self-reconstruction loss $\mathcal{L}_{rec}$, cycle-consistency loss $\mathcal{L}_{cyc}$, and latent constraint $\mathcal{L}_{lat}$:

$$
\mathcal{L}_{topo} = \lambda_{rec}\mathcal{L}_{rec} + \lambda_{cyc}\mathcal{L}_{cyc} + \lambda_{lat}\mathcal{L}_{lat}. \quad (11)
$$

These terms are explicitly formulated as:

$$
\begin{aligned}
\mathcal{L}_{rec} &= \|\hat{I}_{\mathcal{V}} - I_{\mathcal{V}}\|_1 + \|\hat{I}_{\mathcal{I}} - I_{\mathcal{I}}\|_1, \\
\mathcal{L}_{cyc} &= \|\tilde{I}_{\mathcal{V}} - I_{\mathcal{V}}\|_1 + \|\tilde{I}_{\mathcal{I}} - I_{\mathcal{I}}\|_1, \\
\mathcal{L}_{lat} &= \sum_{i=1}^{4}\left(\|\bar{C}_{\mathcal{I}}^i - C_{\mathcal{V}}^i\|_1 + \|\bar{C}_{\mathcal{V}}^i - C_{\mathcal{I}}^i\|_1\right) \\
&\quad + \sum_{i=1}^{4}\left(\|\bar{A}_{\mathcal{V}}^i - A_{\mathcal{V}}^i\|_1 + \|\bar{A}_{\mathcal{I}}^i - A_{\mathcal{I}}^i\|_1\right),
\end{aligned}
\quad (12)
$$

where $\|\cdot\|_1$ denotes the $L_1$ norm, and $i$ represents the scale index. Crucially, $\bar{C}^i$ and $\bar{A}^i$ denote features extracted from the intermediate translations $\bar{I}$ (e.g., $\bar{I}_{\mathcal{I}}$).

**Adversarial Distribution Alignment ($\mathcal{L}_{adv}$).** To guide the generated images to align with target data distributions, we employ discriminators as statistical regularizers. This constraint aggregates the translation loss $\mathcal{L}_{adv}^{trans}$, which guides the intermediate states align with the target modal distribution, and the fusion loss $\mathcal{L}_{adv}^{fus}$, which regularizes the fused result to comply with natural image statistics:

$$
\begin{aligned}
\mathcal{L}_{adv} &= \lambda_{adv}^{trans}\mathcal{L}_{adv}^{trans} + \lambda_{adv}^{fus}\mathcal{L}_{adv}^{fus}, \\
\mathcal{L}_{adv}^{trans} &= \|\mathcal{D}_{\mathcal{I}}(\bar{I}_{\mathcal{I}}) - 1\|_2^2 + \|\mathcal{D}_{\mathcal{V}}(\bar{I}_{\mathcal{V}}) - 1\|_2^2, \\
\mathcal{L}_{adv}^{fus} &= \|\mathcal{D}_{\mathcal{F}}(I_{\mathcal{F}}) - 1\|_2^2,
\end{aligned}
\quad (13)
$$

where $\mathcal{D}_V$, $\mathcal{D}_I$, and $\mathcal{D}_F$ denote the discriminators for visible, infrared, and fused domains, respectively, $1$ represents the real labels, and $\|\cdot\|_2^2$ denotes the MSE loss.

**Manifold Regularization ($\mathcal{L}_{reg}$).** To mitigate structural collapse and semantic drift during stochastic interpolation, we propose $\mathcal{L}_{reg}$ to simultaneously preserve spatial structure in the image domain and regularize attribute trajectories in the latent space, which is formulated as:

$$
\begin{aligned}
\mathcal{L}_{reg} &= \lambda_{fus}\mathcal{L}_{fus} + \lambda_{attr}\mathcal{L}_{attr}, \\
\mathcal{L}_{fus} &= \lambda_{int}\mathcal{L}_{int} + \lambda_{grad}\mathcal{L}_{grad} + \lambda_{ssim}\mathcal{L}_{ssim}, \\
\mathcal{L}_{int} &= \|I_{\mathcal{F}} - I_{\max}\|_1, \mathcal{L}_{grad} = \|\nabla I_{\mathcal{F}} - G_{\max}\|_1, \\
\mathcal{L}_{ssim} &= 1 - \frac{1}{2}\left(\text{SSIM}(I_{\mathcal{F}}, I_{\mathcal{V}}) + \text{SSIM}(I_{\mathcal{F}}, I_{\mathcal{I}})\right), \\
\mathcal{L}_{attr} &= \sum_{i=1}^{4}\left\|\mathcal{P}(A_{\mathcal{F}}^i|A_{\mathcal{I}}^i \to A_{\mathcal{V}}^i) - \alpha\right\|_1,
\end{aligned}
\quad (14)
$$

where $I_{\max} = \max(I_{\mathcal{V}}, I_{\mathcal{I}})$ and $G_{\max} = \max(|\nabla I_{\mathcal{V}}|, |\nabla I_{\mathcal{I}}|)$ denote the pixel-wise intensity and gradient saliency anchors, respectively. Here, $\nabla$ represents the Sobel operator for texture extraction, and SSIM($\cdot$) ([Wang et al., 2004]) enforces local structural consistency. For the latent space, $A_{\mathcal{V}}^i$, $A_{\mathcal{I}}^i$, and $A_{\mathcal{F}}^i$ represent the attribute vectors at scale $i$. The projection operator $\mathcal{P}(A_{\mathcal{F}}^i|A_{\mathcal{I}}^i \to A_{\mathcal{V}}^i)$ calculates the scalar projection ratio of $A_{\mathcal{F}}^i$ onto the latent trajectory defined by the directional vector $(A_{\mathcal{V}}^i - A_{\mathcal{I}}^i)$. Thus, $\mathcal{L}_{fus}$ anchors the spatial structure, while $\mathcal{L}_{attr}$ aligns the latent trajectory with the interpolation factor $\alpha$.

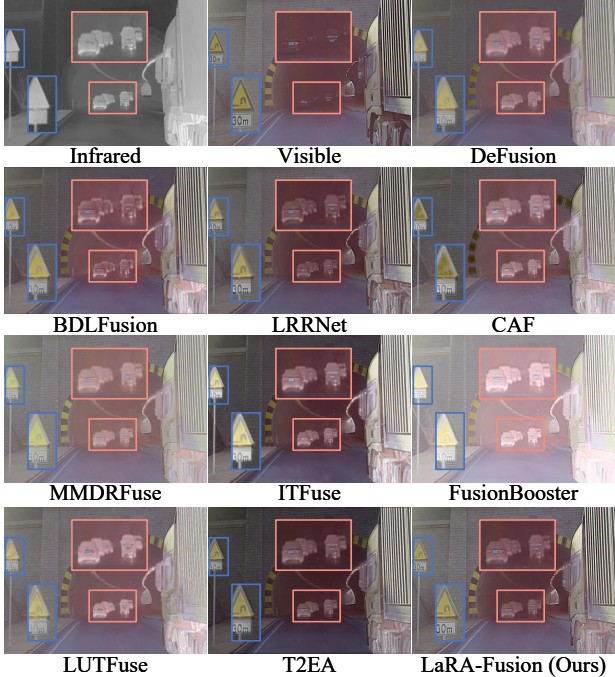

*Figure 4.* Qualitative comparison of LaRA-Fusion with nine state-of-the-art methods on image 02641 from the M3FD dataset.

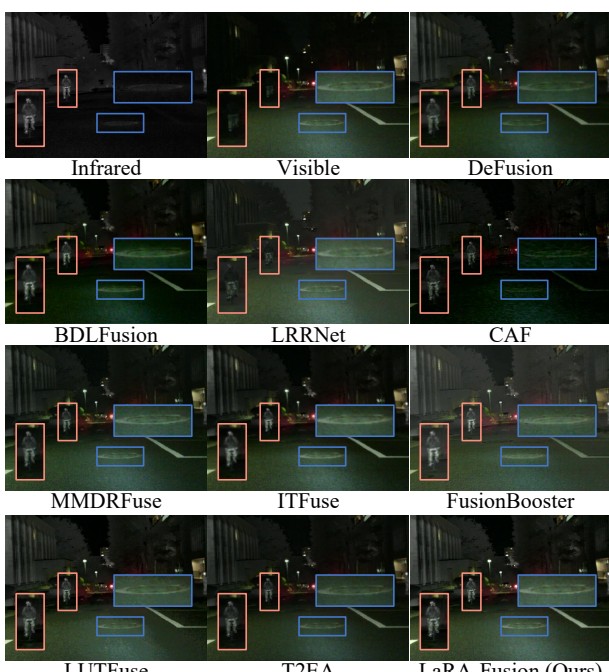

*Figure 5.* Qualitative comparison of LaRA-Fusion with nine state-of-the-art methods on image 01368N from the MSRS dataset.

## 5. Experiment

### 5.1. Experiment Setup

**Datasets.** We train our proposed LaRA-Fusion on 1,083 image pairs from the MSRS dataset (Tang et al., 2022b). Our evaluation involves four public benchmarks: the MSRS test set (300 pairs) for in-domain evaluation, and TNO (Toet & Hogervorst, 2012) (24 pairs), Kaist (Hwang et al., 2015) (150 pairs), and M3FD (Liu et al., 2022) (120 pairs) for cross-dataset evaluation. Notably, we directly apply the model trained on MSRS to the latter three datasets without any fine-tuning to assess generalization capability.

**Optimization.** Our experiments are conducted on a server accelerated by an NVIDIA A800 GPU. During the training phase, we crop the 1,083 image pairs from the MSRS dataset into non-overlapping $128 \times 128$ patches via a sliding window strategy to promote optimization stability and sample diversity. Note that despite this patch-based training scheme, our model supports inputs of arbitrary spatial dimensions during inference. The model is trained for 90 epochs with a batch size of 128. We utilize the Adam optimizer to jointly update the parameters of our generative framework. To accommodate differential training dynamics, the generator is initialized with a learning rate of $5.0 \times 10^{-4}$, whereas the discriminators ($D_\mathcal{V}, D_\mathcal{I}, D_\mathcal{F}$) operate at a reduced rate of $1.0 \times 10^{-5}$. Subsequently, we implement a step-wise decay strategy, where these learning rates are scaled by a factor of 0.8 every 40 epochs to facilitate convergence.

**Hyperparameters.** To balance the contribution of different constraints within our dual-loop mechanism, we empirically configure the weighting hyperparameters as follows: For the topological consistency, we emphasize geometric invertibility by setting $\lambda_{rec} = 4.0$, $\lambda_{cyc} = 4.0$, and $\lambda_{lat} = 3.0$. In the adversarial distribution alignment, we assign equal importance to both translation and fusion objectives with $\lambda_{adv}^{trans} = 1.0$ and $\lambda_{adv}^{fus} = 1.0$. Finally, regarding the manifold regularization, we impose strong structural anchoring by setting $\lambda_{grad} = 5.5$, $\lambda_{int} = 4.0$, and $\lambda_{ssim} = 2.0$, while the attribute alignment weight $\lambda_{attr}$ is set to 1.0 to regularize the attribute manifold. In terms of the architectural configuration, the channel dimensions $C_i$ for the multi-scale stages $i \in \{1, 2, 3, 4\}$ are set to $\{16, 32, 64, 64\}$.

**Evaluation Metrics.** To comprehensively assess the performance of our method across four IVIF datasets, we conduct comparative experiments against nine representative state-of-the-art (SOTA) methods: DeFusion (Liang et al., 2022), BDLFusion (Liu et al., 2023), LRRNet (Li et al., 2023a), CAF (Liu et al., 2024), MMDRFuse (Deng et al., 2024), IT-Fuse (Tang et al., 2024), FusionBooster (Cheng et al., 2025), LUTFuse (Yi et al., 2025), and T2EA (Huang et al., 2025b). Quantitatively, we employ a suite of six established metrics: Entropy (EN), Standard Deviation (SD), Average Gradient (AG), Mutual Information (MI), Visual Information Fidelity (VIF), and Edge-based Quality ($Q_{abf}$). Consistent with the standards defined in (Ma et al., 2019a), higher scores in these metrics indicate superior fusion performance.

*Table 1.* Quantitative comparison on different IVIF datasets, with the best and second-best values in **bold** and underline

| Method | Source | Kaist | | | | | | M3FD | | | | | |
|---|---|---|---|---|---|---|---|---|---|---|---|---|---|
| | | EN | SD | AG | MI | VIF | $Q_{abf}$ | EN | SD | AG | MI | VIF | $Q_{abf}$ |
| DeFusion | ECCV'22 | 6.74 | 43.58 | 2.26 | 2.58 | 0.72 | 0.50 | 6.56 | 29.13 | 2.79 | 2.07 | 0.56 | 0.34 |
| BDLFusion | IJCAI'23 | 6.99 | 43.05 | 2.62 | 2.37 | 0.85 | 0.57 | 6.77 | 31.02 | 3.20 | 2.05 | 0.65 | 0.45 |
| LRRNet | TPAMI'23 | 6.96 | 44.19 | 2.83 | 2.11 | 0.73 | 0.49 | 6.55 | 29.27 | 3.81 | 2.02 | 0.57 | 0.51 |
| CAF | IJCAI'24 | 6.52 | 30.69 | 2.79 | 1.83 | 0.65 | 0.47 | 6.92 | 35.67 | 4.65 | 1.98 | 0.61 | 0.53 |
| MMDRFuse | ACMMM'24 | 6.89 | 42.55 | 2.85 | 2.83 | 0.84 | 0.63 | 6.58 | 28.47 | 3.61 | 2.35 | 0.67 | 0.52 |
| ITFuse | PR'24 | 7.05 | 42.86 | 2.60 | 2.26 | 0.74 | 0.51 | 6.83 | 35.97 | 3.15 | 1.99 | 0.56 | 0.37 |
| FusionBooster | IJCV'25 | 6.84 | 33.35 | 2.77 | 1.83 | 0.69 | 0.44 | 6.87 | 33.34 | 3.84 | 1.98 | 0.53 | 0.41 |
| LUTFuse | ICCV'25 | 7.01 | 53.37 | 3.43 | 3.38 | 0.83 | 0.63 | 6.85 | 36.02 | 4.38 | 3.07 | 0.65 | 0.53 |
| T2EA | TCSVT'25 | 6.91 | 46.80 | 2.88 | 2.60 | 0.81 | 0.63 | 6.68 | 30.19 | 3.89 | 2.04 | 0.60 | 0.52 |
| **LaRA-Fusion** | This Paper | **7.14** | **59.67** | **3.58** | **3.53** | **0.97** | **0.70** | **7.02** | **41.67** | **5.33** | **3.28** | **0.86** | **0.65** |

| Method | Source | MSRS | | | | | | TNO | | | | | |
|---|---|---|---|---|---|---|---|---|---|---|---|---|---|
| | | EN | SD | AG | MI | VIF | $Q_{abf}$ | EN | SD | AG | MI | VIF | $Q_{abf}$ |
| DeFusion | ECCV'22 | 6.31 | 34.43 | 2.55 | 2.14 | 0.75 | 0.51 | 6.69 | 31.91 | 2.89 | 1.76 | 0.53 | 0.34 |
| BDLFusion | IJCAI'23 | 6.05 | 33.05 | 2.54 | 1.85 | 0.73 | 0.49 | 6.99 | 38.18 | 3.60 | 1.64 | 0.59 | 0.41 |
| LRRNet | TPAMI'23 | 6.15 | 31.16 | 2.59 | 2.00 | 0.54 | 0.45 | 7.07 | 41.99 | 4.12 | 1.71 | 0.55 | 0.35 |
| CAF | IJCAI'24 | 5.83 | 26.11 | 3.09 | 1.30 | 0.59 | 0.43 | 6.92 | 36.54 | 4.41 | 1.70 | 0.57 | 0.43 |
| MMDRFuse | ACMMM'24 | 6.44 | 36.88 | 3.15 | 2.50 | 0.85 | 0.59 | 6.57 | 30.51 | 3.30 | 1.83 | 0.59 | 0.40 |
| ITFuse | PR'24 | 6.38 | 34.77 | 2.42 | 1.82 | 0.67 | 0.41 | 7.02 | 39.93 | 3.23 | 1.49 | 0.52 | 0.34 |
| FusionBooster | IJCV'25 | 6.29 | 29.82 | 2.99 | 1.43 | 0.64 | 0.42 | 6.68 | 32.66 | 3.88 | 1.59 | 0.47 | 0.31 |
| LUTFuse | ICCV'25 | 6.64 | 42.15 | 3.69 | 2.92 | 0.93 | 0.63 | 6.97 | 41.09 | 3.96 | 2.50 | 0.63 | 0.46 |
| T2EA | TCSVT'25 | 6.22 | 32.57 | 2.72 | 2.20 | 0.77 | 0.55 | 7.02 | 39.47 | 4.14 | 1.76 | 0.62 | 0.46 |
| **LaRA-Fusion** | This Paper | **6.77** | **45.73** | **3.76** | **2.98** | **1.03** | **0.66** | **7.10** | **50.95** | **4.90** | **2.52** | **0.82** | **0.49** |

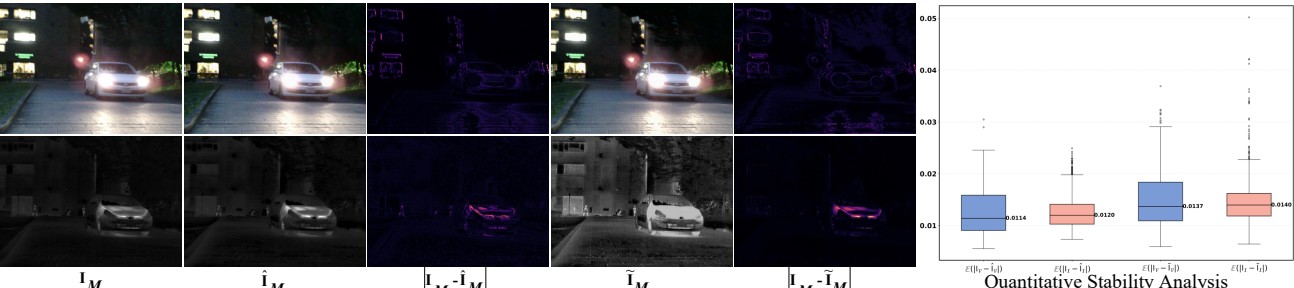

$$I_M \qquad \hat{I}_M \qquad |I_M - \hat{I}_M| \qquad \tilde{I}_M \qquad |I_M - \tilde{I}_M| \qquad \text{Quantitative Stability Analysis}$$

*Figure 6.* **Reconstruction Fidelity and Stability.** Left: Visual comparisons and spatial residual heatmaps for direct and cyclic reconstruction pipelines. Right: Quantitative MAE distribution statistics across the MSRS training set.

## 5.2. Comparative Evaluation

**Qualitative Comparisons.** To facilitate visual inspection, we colorize the fused results by adopting the chrominance channels from the source visible images. Qualitative comparisons are presented for a nighttime scenario in Figure 5 and a daytime scenario in Figure 4. In the daytime scene, LaRA-Fusion yields a perceptually natural outcome. In contrast, several comparative methods inject excessive thermal radiation into the large truck region. Consequently, this results in unnatural intensity distributions given the well-lit conditions. Our proposed approach effectively suppresses these artifacts and preserves the semantic appearance of

the object. Furthermore, the traffic sign in the blue box exhibits sharper edges and favorable legibility. Meanwhile, the thermal saliency of the vehicle inside the tunnel (coral box) is well-preserved without compromising global contrast. Shifting our focus towards the nighttime scenario in Figure 5 from MSRS dataset, LaRA-Fusion continues to demonstrate robust integration capabilities. The thermal signature highlighted in the coral box remains prominent, preserving the visibility of salient heat sources. Simultaneously, the urban manhole cover within the blue box retains sharp structural details, closely mirroring the intricate texture information originating from the visible source rather than being obscured by thermal crossover.

*Table 2.* Ablation studies on MSRS dataset

|            | EN   | SD    | AG   | MI   | VIF  | $Q_{abf}$ |
|------------|------|-------|------|------|------|-----------|
| w/o Inner  | 6.47 | 43.18 | 3.68 | 2.68 | 0.97 | 0.63      |
| w/o Outer  | 6.58 | 43.25 | 3.61 | 2.87 | 0.96 | 0.64      |
| w/o MRB    | 6.69 | 43.26 | 3.56 | 2.95 | 0.99 | 0.61      |
| w/o HAA    | 6.63 | 43.57 | 3.54 | 2.98 | 0.93 | 0.64      |
| w/o Stoch. | 6.70 | 43.37 | 3.57 | 2.93 | 1.00 | 0.65      |
| Ours       | **6.77** | **45.73** | **3.76** | **2.98** | **1.03** | **0.66** |

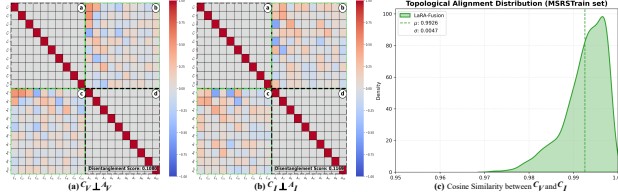

*Figure 7.* **Disentanglement and Alignment.** The minimal correlations indicate that content and attribute features are effectively disentangled, while the high similarity density demonstrates our method preserves consistent latent topology across modalities.

**Quantitative Comparisons.** As presented in Table 1, the proposed LaRA-Fusion consistently achieves competitive performance across all six metrics on four IVIF datasets. Specifically, the top-ranking MI and EN scores indicate that our method effectively preserves the rich information contained in the source images. In terms of visual quality, the superior SD and VIF values suggest that the fused results maintain high contrast and align well with human visual perception. Furthermore, the leading AG and $Q_{abf}$ results demonstrate the method's capability in retaining fine gradients and structural edge details. These consistent improvements across diverse metrics and datasets validate the effectiveness and generalization ability of our framework compared to other state-of-the-art approaches.

### 5.3. Ablation Study

To evaluate the efficacy of LaRA-Fusion, we conducted a systematic ablation study on the MSRS dataset (Table 2), isolating specific components to dissect their contributions toward robust feature learning. First, regarding the Dual-Loop Constraints, removing the Inner Loop forfeits geometric reversibility, causing the encoder to discard fine-grained structural details essential for restoration, as evidenced by declines in MI and SD. Conversely, eliminating the adversarial alignment imposed by the Outer Loop causes generated representations to deviate from the target data distribution, leading to textural degradation, which directly compromises the VIF metric reflecting perceptual fidelity.

We further systematically assessed the contribution of our specialized designs by substituting them with baseline counterparts. Replacing the MRB with a standard convolutional layer eliminates residual channel aggregation, which impairs the model's ability to alleviate topological gaps and reduces edge preservation ($Q_{abf}$). Similarly, substituting the dynamic HAA with static non-linear projection layers removes the adaptive attribute arbitration, resulting in unbalanced feature integration. Finally, we analyzed the impact of Stochastic Exploration by fixing $\alpha$ and nullifying the regularization ($L_{attr} = 0$). This removal confines the encoder to deterministic mappings, encouraging the learning of degenerate numerical shortcuts rather than domain-invariant structures, which consequently limits generalization.

### 5.4. Interpretability and Robustness Analysis

**Latent Robustness.** To validate encoding stability, we analyze the reconstruction fidelity. Figure 6 visualizes the reconstruction pipeline and the corresponding spatial residual maps. The absence of structural artifacts in these maps indicates that the encoder captures semantic features rather than relying on degenerate numerical shortcuts. Quantitatively, the MSRS training set statistics demonstrate a compact error distribution, with the median MAE consistently remaining below 0.015 across generative pathways. These results suggest that our dual-loop mechanism enforces effective manifold constraints, promoting reliable latent representations.

**Latent Topology and disentanglement.** We further probe the internal mechanism in Figure 7 to validate the robustness of the latent space. First, the decoupling analysis reveals that the content-attribute correlation (specifically quadrants ⓑ and ⓒ within Figure 7 (a)-(b)) remains negligible (below 0.12). This quasi-independence indicates that geometric structures are decoupled from attributes, effectively mitigating the "leakage" path for degenerate shortcuts. Second, the cosine similarity KDE demonstrates consistent topological alignment. This result indicates that different modalities are mapped into a unified geometric space, remaining structurally aligned yet effectively disentangled.

## 6. Conclusion

In this paper, we introduce LaRA-Fusion, a latent-robust adaptation framework designed to address the persistent challenges of topological inconsistency and degenerate numerical shortcuts prevalent in unsupervised image fusion. To mitigate these optimization pitfalls, LaRA-Fusion constructs a dual-loop mechanism that interlocks geometric invertibility with adversarial alignment. Specifically, MRB, HAA, and MSB synergize to alleviate latent misalignments and modulate synthesis, while stochastic attribute interpolation promotes structural stability under environmental shifts. Extensive experiments confirm that LaRA-Fusion performs favorably against state-of-the-art methods, demonstrating that enforcing topological integrity is crucial for generating robust, semantically aligned fused imagery.

## Acknowledgment

This work was supported in part by the National Key Research and Development Plan of China under Grant 2021YFB3600503, in part by the National Natural Science Foundation of China under Grant 61972097 and U21A20472, in part by the Major Scientific Research Project for Technology Promotes Police under Grant 2025YZ040003, 2024YZ040001, in part by the Natural Science Foundation of Fujian Province under Grant 2025J01536, 2025J01541.

## Impact Statement

This paper presents work whose goal is to advance the field of Machine Learning. There are many potential societal consequences of our work, none which we feel must be specifically highlighted here.

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

# A. Discriminator

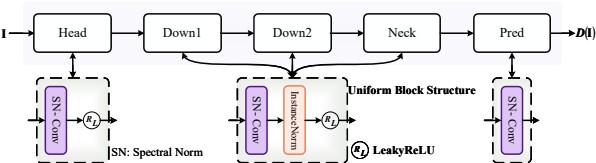

*Figure 8.* **Architecture of the Discriminator.** The network maps the input image I to a validity map $\mathcal{D}(I)$ via a hierarchical progression comprising a **Head**, cascaded downsampling stages (**Down1**, **Down2**), and a **Neck** bottleneck, finally culminating in the **Pred** layer. The uniform block structure details the shared topology employed by the Down and Neck modules. Crucially, spectral normalization is consistently applied to the convolutional layers throughout the network to enforce Lipschitz continuity, promoting stable distribution alignment.

**Discriminator Structure.** As illustrated in Figure 8, we employ a Markovian discriminator architecture, commonly referred to as PatchGAN (Isola et al., 2017), to classify local image patches rather than the entire image globally. This design guides the model to attend to high-frequency structural details, inhibiting the generator from relying on degenerate numerical shortcuts that manifest as blurry textures. The network maps the input image I to a validity map $\mathcal{D}(I)$ through a hierarchical pathway comprising *Head*, *Down1*, *Down2*, *Neck*, and *Pred* modules. To stabilize adversarial training, we apply spectral normalization (Miyato et al., 2018) to all convolutional layers. The core component, denoted as the uniform block structure, consists of a spectrally normalized $4 \times 4$ convolution, followed by Instance Normalization and a LeakyReLU activation (slope=0.2). Specifically, the *Head* initiates feature extraction (without normalization) and downsamples the resolution. *Down1* and *Down2* progressively deepen the channels ($64 \rightarrow 128 \rightarrow 256$) while reducing spatial dimensions to aggregate semantic context. The *Neck* layer further expands the feature depth to 512 channels with a stride of 1. Finally, the *Pred* layer projects the features onto a single-channel decision map, providing spatially localized adversarial feedback.

**Discriminator Optimization Objectives.** To stabilize the adversarial training and mitigate the vanishing gradient problem, we adopt the Least Squares GAN (LSGAN) formulation (Mao et al., 2017). The discriminators facilitate distribution alignment, distinguishing between the generated samples and the real data distribution.

**1) Translation Discriminators ($\mathcal{D}_\mathcal{V}, \mathcal{D}_\mathcal{I}$).** The modality-specific discriminators $\mathcal{D}_\mathcal{V}$ and $\mathcal{D}_\mathcal{I}$ are designed to align the cross-modal translations ($\bar{I}_\mathcal{V}, \bar{I}_\mathcal{I}$) with the textural statistics of the target domain. They minimize the discrepancy between the synthesized pseudo-images and the real source

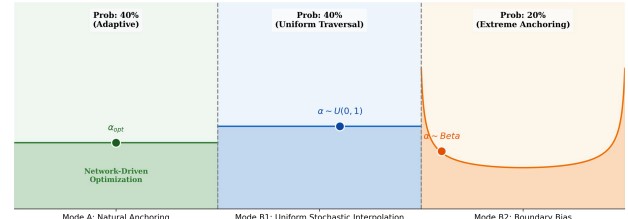

*Figure 9.* **Visualization of the hybrid manifold exploration strategy.** The training process stochastically alternates between **Natural Anchoring** for optimal parameter learning, **Uniform Stochastic Interpolation** for continuous latent traversal, and **Boundary Bias** using a Beta distribution to enhance robustness against extreme attribute shifts.

images. Formally:

$$
\begin{aligned}
\mathcal{L}_{\mathcal{D}_\mathcal{V}} &= \mathbb{E}\left[(\mathcal{D}_\mathcal{V}(I_\mathcal{V}) - 1)^2\right] + \mathbb{E}\left[(\mathcal{D}_\mathcal{V}(\bar{I}_\mathcal{V}))^2\right], \\
\mathcal{L}_{\mathcal{D}_\mathcal{I}} &= \mathbb{E}\left[(\mathcal{D}_\mathcal{I}(I_\mathcal{I}) - 1)^2\right] + \mathbb{E}\left[(\mathcal{D}_\mathcal{I}(\bar{I}_\mathcal{I}))^2\right],
\end{aligned}
\tag{15}
$$

where $\mathbb{E}$ denotes the expectation operation. This constraint inhibits the generator from producing out-of-distribution artifacts that satisfy numerical reduction but violate domain appearance laws.

**2) Fusion Discriminator ($\mathcal{D}_\mathcal{F}$).** Unlike the translation branch, the fusion discriminator $\mathcal{D}_\mathcal{F}$ aims to validate the "naturalness" of the fused result $I_\mathcal{F}$. Since a high-quality fused image should align with the joint distribution of natural scenes encompassing both spectra, we treat both source images $I_\mathcal{V}$ and $I_\mathcal{I}$ as real samples. Formally:

$$
\mathcal{L}_{\mathcal{D}_\mathcal{F}} = \underbrace{\frac{1}{2}\left(\mathbb{E}\left[(\mathcal{D}_\mathcal{F}(I_\mathcal{V}) - 1)^2\right] + \mathbb{E}\left[(\mathcal{D}_\mathcal{F}(I_\mathcal{I}) - 1)^2\right]\right)}_{\text{Real Naturalness Prior}}
$$
$$
+ \underbrace{\mathbb{E}\left[(\mathcal{D}_\mathcal{F}(I_\mathcal{F}))^2\right]}_{\text{Fake Penalty}}.
\tag{16}
$$

This dual-anchor strategy provides an effective regularization constraint, conditioning the fused output to exhibit realistic gradient continuity and intensity distributions consistent with real sensor data, thereby suppressing degenerate solutions that rely on high-frequency noise stitching.

# B. Hybrid Manifold Exploration Strategy

To promote the topological continuity of the latent space and mitigate degenerate numerical shortcuts, we implement a **hybrid manifold exploration strategy**. As visualized in Figure 9, this strategy modulates the attribute interpolation factor $\alpha$ via stochastic arbitration during training, governed by a probabilistic mechanism divided into three distinct modes. First, the **Natural Anchoring** mode (40%) is adopted, where the attribute arbitration network autonomously determines the optimal fusion weights without external intervention. This stabilizes the optimization

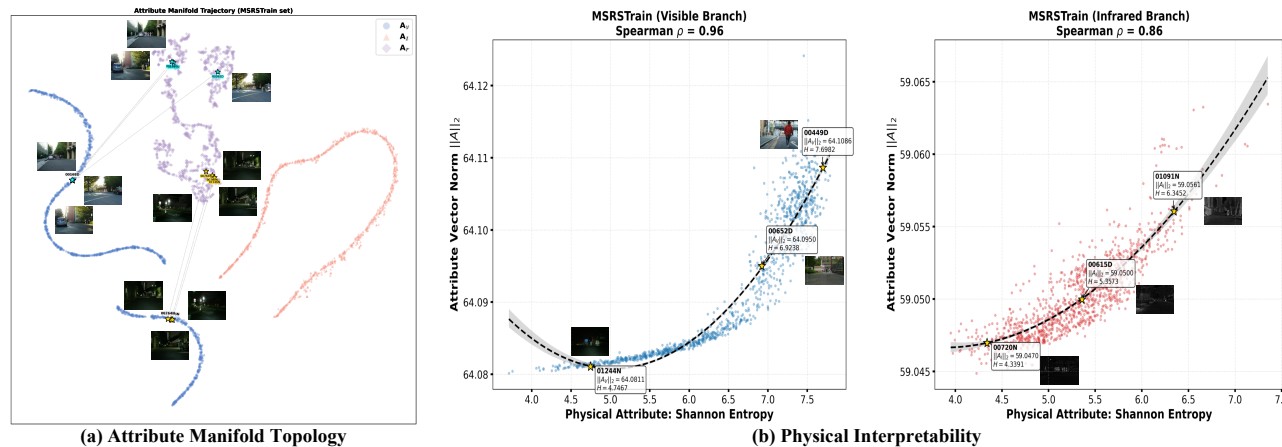

**(a) Attribute Manifold Topology**  **(b) Physical Interpretability**

*Figure 10.* **Analysis of Latent Manifold Topology and Physical Grounding. (a) Attribute Manifold Topology.** The t-SNE visualization reveals a clear and structured latent manifold. The infrared attributes (red) form a smooth trajectory, while the visible attributes (blue) exhibit a semantic discontinuity, distinctly clustering into daytime and nighttime domains rather than a chaotic distribution typical of degenerate shortcuts. **(b) Physical Interpretability.** We observe a strong monotonic correlation between the attribute vector norm $\|A\|_2$ and the physical Shannon entropy ($H$) of the input scenes. High Spearman correlations confirm that the encoder captures intrinsic physical complexity rather than embedding imperceptible high-frequency noise.

*Table 3.* Representative anchors sampled from the regression manifold. Note the strict ascending order of $\|A\|_2$ corresponding to the physical entropy.

| Modality | Stratum | Sample ID | Entropy ($H$) | Vector Norm $\|A\|_2$ |
|---|---|---|---|---|
| Visible | Low | 01244N | 4.7467 | 64.0811 |
| Visible | Medium | 00625D | 6.9238 | 64.0950 |
| Visible | High | 00449D | 7.6982 | 64.1086 |
| Infrared | Low | 00720N | 4.3391 | 59.0470 |
| Infrared | Medium | 00615D | 5.3573 | 59.0500 |
| Infrared | High | 01091N | 6.3452 | 59.0561 |

trajectory, facilitating effective convergence to a physically plausible manifold center. To counteract the formation of latent voids, we apply **Uniform Sampling** (40%) by sampling $\alpha$ from the uniform distribution $U(0, 1)$. This drives the decoder to generalize across continuous intermediate states, enabling smooth cross-modal interpolation. Finally, to comprehensively evaluate structural integrity, we introduce a **Boundary Bias** (20%) by sampling $\alpha$ from a Beta distribution $B(0.5, 0.5)$. This concentrates probability density at the manifold extremities, inducing the content encoder to maintain robust semantic representations invariant to extreme attribute shifts, thereby inhibiting the model from collapsing into trivial identity mappings.

## C. Manifold Topology and Physical Interpretability

To verify that LaRA-Fusion learns a topologically meaningful latent space rather than relying on degenerate numerical shortcuts, we investigate the fundamental manifold geometry and physical interpretability. This serves as the founda-

tion for verifying the subsequent structural disentanglement and robustness properties.

**Topological Continuity of Attribute Manifold.** Figure 10(a) visualizes the t-SNE projection of the learned attribute vectors on the MSRS training set, revealing an organized manifold structure that adheres to the intrinsic data distribution. Specifically, the infrared attributes (red) form a continuous trajectory, reflecting the stable thermal continuum across diverse scenes. In contrast, the visible attributes (blue) display a structured topological separation into two distinct clusters. Visual inspection of the anchor points indicates that this gap corresponds to the semantic domain shift between daytime and nighttime scenarios. This coherent clustering suggests that the encoder disentangles meaningful environmental attributes, effectively avoiding the disordered or uniform distributions typically associated with noise-based shortcut learning.

**Physical Interpretability of Latent Vectors.** To further validate that the learned representations are grounded in physical reality, we investigate the relationship between the magnitude of the attribute vector ($\|A\|_2$) and the information density of the source image, quantified by Shannon entropy. As illustrated in Figure 10(b), the scatter plots reveal a robust monotonic relationship, substantiated by high Spearman correlation coefficients of 0.96 for the visible branch and 0.86 for the infrared branch. Rather than exhibiting random fluctuations, the attribute encoder yields vector magnitudes that follow the variations in scene information. This behavior supports the premise that the latent space is continuous and physically interpretable, rather than being driven by random noise.

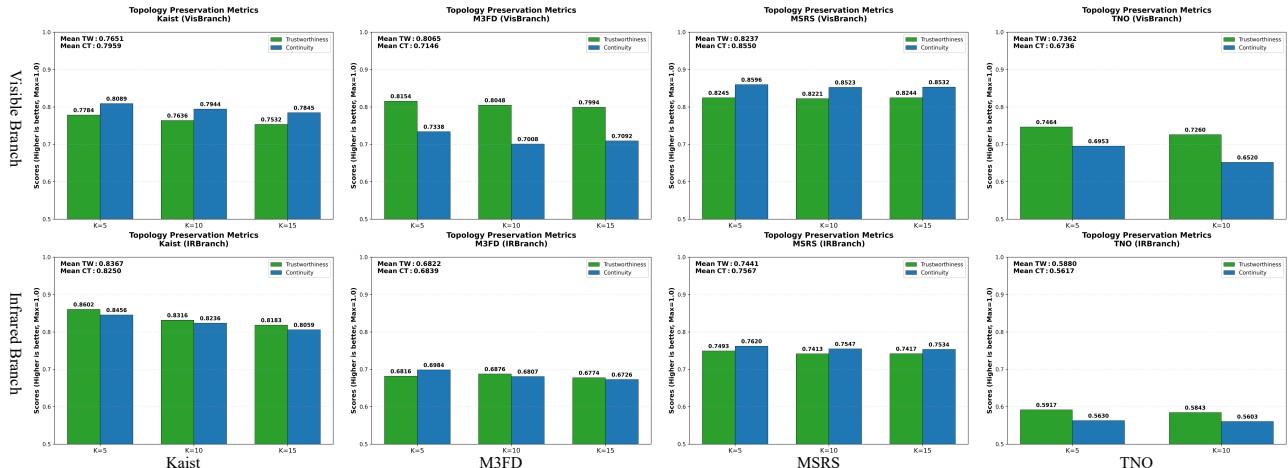

*Figure 11.* **Evaluation of Topology Preservation Metrics.** The bar charts illustrate the Trustworthiness and Continuity scores across the Kaist, M3FD, MSRS, and TNO datasets (from left to right) for both source modalities at varying neighborhood sizes. The consistently high values indicate that local and global geometric structures are reliably maintained during latent projection.

*Table 4.* **Quantitative Assessment of Latent Manifold Preservation.** Mean Trustworthiness and Continuity scores for both source modalities across multiple datasets, demonstrating the model's capacity to preserve intrinsic topological relationships without severe spatial distortion.

| Dataset | Trustworthiness | | Continuity | |
|---|---|---|---|---|
| | Visible($\uparrow$) | Infrared($\uparrow$) | Visible($\uparrow$) | Infrared($\uparrow$) |
| Kaist | 0.7651 | 0.8367 | 0.7959 | 0.8250 |
| M3FD | 0.8065 | 0.6822 | 0.7146 | 0.6839 |
| MSRS | 0.8237 | 0.7441 | 0.8550 | 0.7567 |
| TNO | 0.7362 | 0.5880 | 0.6736 | 0.5617 |

To explicitly visualize this sensitivity, we sampled representative anchors along the regression manifold as detailed in Table 3. Observations demonstrate that the model effectively captures saliency, mapping the progression from illumination-constrained or low-contrast inputs (e.g., 01244N, 00720N) to texture-rich environments with prominent targets (e.g., 00449D, 01091N). This alignment indicates that the learned latent space correlates well with physical characteristics, distinguishing variations in visual complexity by assigning ascending vector norms within a narrow numerical range.

## D. Topological Consistency and Manifold Preservation

Building upon the qualitative manifold visualizations and physical interpretability established previously, we proceed to quantify the topological consistency of the latent space. While t-SNE projections effectively illustrate local neighborhood structures, they can sometimes distort global geometric relationships. To provide a more robust assessment, we evaluate two standard manifold learning metrics: Trustwor-

thiness ($TW$) and Continuity ($CT$) (Venna & Kaski, 2006). These metrics measure the extent to which $k$-nearest neighbor rankings are preserved during the encoding process, mapping from the original input space to the disentangled latent representations (Moor et al., 2020; Chen et al., 2022).

Mathematically, these metrics are formulated based on local rank discrepancies as follows:

$$TW(k) = 1 - C \sum_{i=1}^{N} \sum_{j \in U_i^{(k)}} (r(i,j) - k)$$

$$CT(k) = 1 - C \sum_{i=1}^{N} \sum_{j \in V_i^{(k)}} (\hat{r}(i,j) - k)$$

where $r(i,j)$ and $\hat{r}(i,j)$ represent the ranks of the data points, while $U_i^{(k)}$ and $V_i^{(k)}$ denote the sets of points that are among the $k$ nearest neighbors in the latent space but not in the original space, and vice versa. Additionally, $C$ is a normalization factor based on the maximum possible rank error to scale the metrics into the $[0,1]$ range.

As illustrated in Figure 11 and detailed in Table 4, the results across the Kaist, M3FD, MSRS, and TNO datasets consistently demonstrate high scores. For instance, the MSRS dataset yields a mean Trustworthiness of 0.8237 and a Continuity of 0.8550 in the visible branch. These metrics indicate that the encoding process preserves intrinsic geometric relationships without inducing severe spatial distortions, providing quantitative support for the topology-related claims.

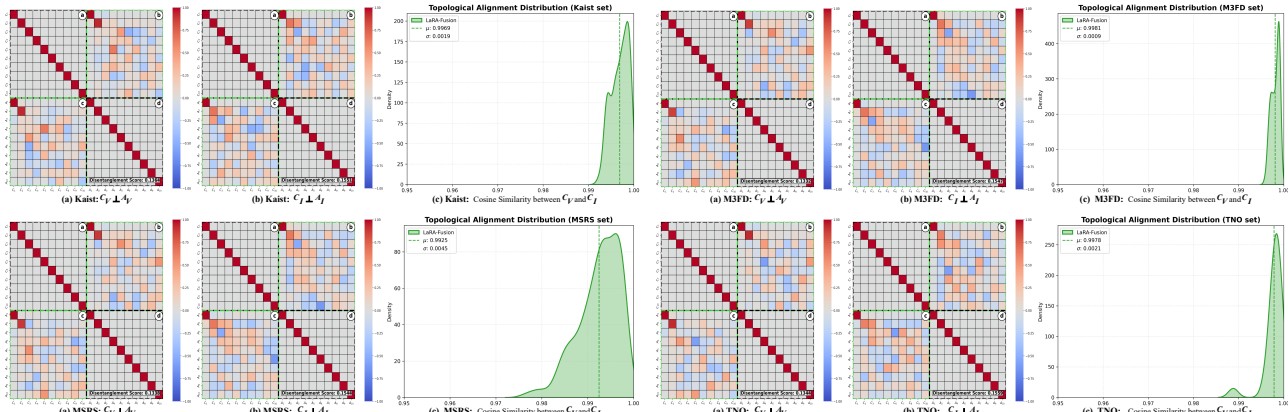

*Figure 12.* **Analysis of Structural Disentanglement and Topological Alignment across four datasets** (Top-Left: Kaist, Top-Right: M3FD, Bottom-Left: MSRS, Bottom-Right: TNO). The Pearson correlation matrices between content $C$ and attribute $A$ vectors exhibit consistently low off-diagonal correlations (e.g., Kaist: $\rho_{\mathcal{V}} \approx 0.136$, $\rho_{\mathcal{I}} \approx 0.155$), indicating statistical independence. The Kernel Density Estimation (KDE) of cosine similarity between $C_{\mathcal{V}}$ and $C_{\mathcal{I}}$ shows sharp concentrations near unity (e.g., M3FD Mean: 0.9981), validating the extraction of a modality-invariant geometric basis.

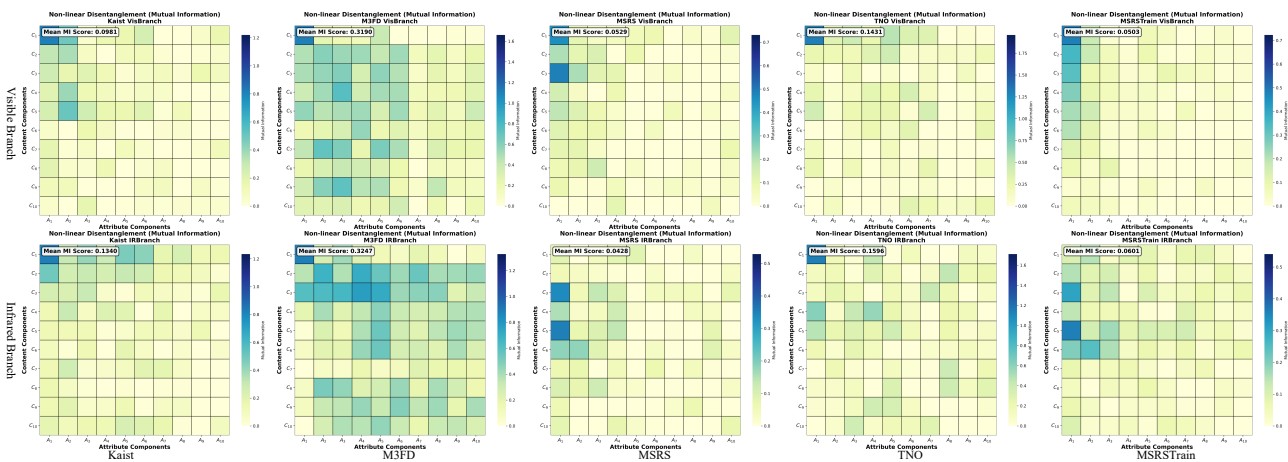

*Figure 13.* **Latent Mutual Information (LMI) Analysis.** Evaluation of non-linear statistical dependence between content $C$ and attribute $A$ representations across different datasets (Top: Visible $\mathcal{V}$, Bottom: Infrared $\mathcal{I}$). The consistently suppressed LMI values suggest that the representations remain statistically independent, supporting structural decoupling under continuous manifold variations.

*Table 5.* **Quantitative Analysis of Latent Space Topology.** We evaluate the performance in terms of feature decoupling (including LMI) and latent alignment. The consistent performance across all datasets confirms the stability of our proposed method.

| Dataset | Decoupling | | LMI | | Alignment | |
|---|---|---|---|---|---|---|
| | $\mathcal{V}(\downarrow)$ | $\mathcal{I}(\downarrow)$ | $\mathcal{V}(\downarrow)$ | $\mathcal{I}(\downarrow)$ | **Mean** $(\mu \uparrow)$ | **Std** $(\sigma \downarrow)$ |
| Kaist | 0.1364 | 0.1551 | 0.0981 | 0.1340 | 0.9969 | 0.0019 |
| M3FD | 0.1332 | 0.1547 | 0.3190 | 0.3247 | 0.9981 | 0.0009 |
| MSRS | 0.1338 | 0.1544 | 0.0529 | 0.0428 | 0.9925 | 0.0045 |
| TNO | 0.1344 | 0.1559 | 0.1431 | 0.1596 | 0.9978 | 0.0021 |

# E. Structural Disentanglement and Topological Alignment

Building upon the physical interpretability and quantitative verification of topological consistency, we further examine

the structural integrity of the learned latent space. A robust fusion framework is typically expected to satisfy two critical topological conditions: **Decoupling**, which implies that content and attributes are statistically independent to minimize information leakage, and **Alignment**, which ensures that heterogeneous inputs approximate a shared geometric basis. Figure 12, Figure 13 and Table 5 present a comprehensive evaluation of these properties across different datasets.

**Decoupling of Disentangled Representations.** To verify that the encoder avoids embedding modality-specific details into the content code, we analyze the statistical dependence between the extracted content $C$ and attribute $A$ vectors. The Pearson correlation (Benesty et al., 2009) coefficient matrices for both visible ($\mathcal{V}$) and infrared ($\mathcal{I}$) modalities demonstrate that the off-diagonal correlations (specifically quadrants ⓑ and ⓒ within each subfigure of Figure 12 (a)-

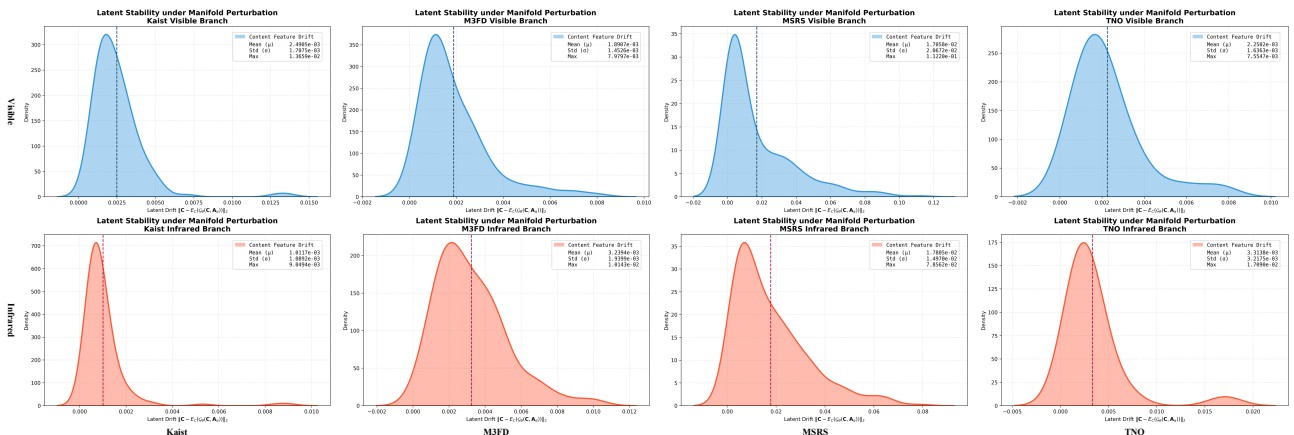

*Figure 14.* **Latent Stability under Stochastic Manifold Shift.** KDE plots of feature drift across Kaist, M3FD, MSRS, and TNO. Drift is measured as $\|\boldsymbol{C} - E_C(\mathcal{G}_\theta(\boldsymbol{C}, \boldsymbol{A}_\alpha))\|_2$, where $\alpha$ denotes stochastic attribute variation. The sharp concentration near zero (e.g., 0.0010 for Kaist infrared branch) indicates that content representations remain structurally stable despite significant manifold shifts.

(b)) remain negligible. Quantitatively, we define a Decoupling Score based on the average absolute cross-correlation:

$$Score = \frac{1}{K^2} \sum_{i=1}^{K} \sum_{j=1}^{K} |\text{Corr}(\boldsymbol{C}_i, \boldsymbol{A}_j)| \quad (17)$$

where $K$ denotes the total number of feature dimensions, and $\text{Corr}(\cdot, \cdot)$ represents the Pearson correlation coefficient between the $i$-th dimension of $\boldsymbol{C}$ and the $j$-th dimension of $\boldsymbol{A}$. As detailed in Table 5, these scores are consistently low, with the Kaist dataset yielding interaction scores of approximately 0.1364 ($\mathcal{V}$) and 0.1551 for source modalities.

Furthermore, since low linear correlation does not necessarily guarantee complete decoupling, we evaluate the Latent Mutual Information (LMI) (Belghazi et al., 2018) to measure non-linear dependence between the branches:

$$LMI = \frac{1}{K^2} \sum_{i=1}^{K} \sum_{j=1}^{K} I(\boldsymbol{C}_i; \boldsymbol{A}_j) \quad (18)$$

where $I(\cdot; \cdot)$ denotes the mutual information (Shannon, 1948) between the $i$-th and $j$-th principal components of $\boldsymbol{C}$ and $\boldsymbol{A}$. Our evaluations reveal that LMI remains consistently low across the tested scenarios. For instance, the MSRS dataset records low LMI values (0.0529 and 0.0428) for the different source modalities, indicating minimal information overlap. These consistently low values suggest that $\boldsymbol{C}$ and $\boldsymbol{A}$ likely occupy statistically independent subspaces, thereby supporting the premise that LaRA-Fusion effectively decouples geometric structure from transient modality attributes.

**Topological Consistency of Content Anchors.** A core assumption of the dual-loop constraint is that the geometric topology of a scene should remain consistent across modalities. We quantify this by evaluating the cosine similarity between the content features extracted from paired inputs, denoted as $\text{sim}(\boldsymbol{C}_\mathcal{V}, \boldsymbol{C}_\mathcal{T})$. The Kernel Density Estimation (KDE) (Silverman, 2018) plots reveal that these similarity scores are sharply concentrated near unity across all datasets. Specifically, referencing the topological statistics in Table 5, the mean alignment scores achieve 0.9969 on Kaist, 0.9981 on M3FD, 0.9925 on MSRS and 0.9978 on TNO. Furthermore, the low standard deviations ($\sigma < 0.005$) support the stability of this alignment. Such strong alignment suggests that the content encoder filters out spectral disparities, anchoring the representations to a unified, modality-agnostic manifold. This strong topological alignment provides a stable structural foundation for the subsequent fusion stage.

## F. Latent Space Robustness under Stochastic Manifold Exploration

While the previous structural analysis indicates the static independence of the representations, a resilient fusion framework should also demonstrate dynamic stability against attribute variability. To verify that the content anchor $\boldsymbol{C}$ captures intrinsic geometric invariants rather than fragile numerical correlations, we introduce a stress test based on stochastic attribute variation. Distinct from the interpolation strategy used during training, we employ a broad stochastic sampling protocol during inference to simulate diverse stylistic variations throughout the attribute manifold. Specifically, we systematically modulate the attribute vector $\boldsymbol{A}$ by scaling its magnitude and injecting Gaussian noise across varying intensity levels. We then measure the resulting displacement in the content space after a full generation-encoding cycle.

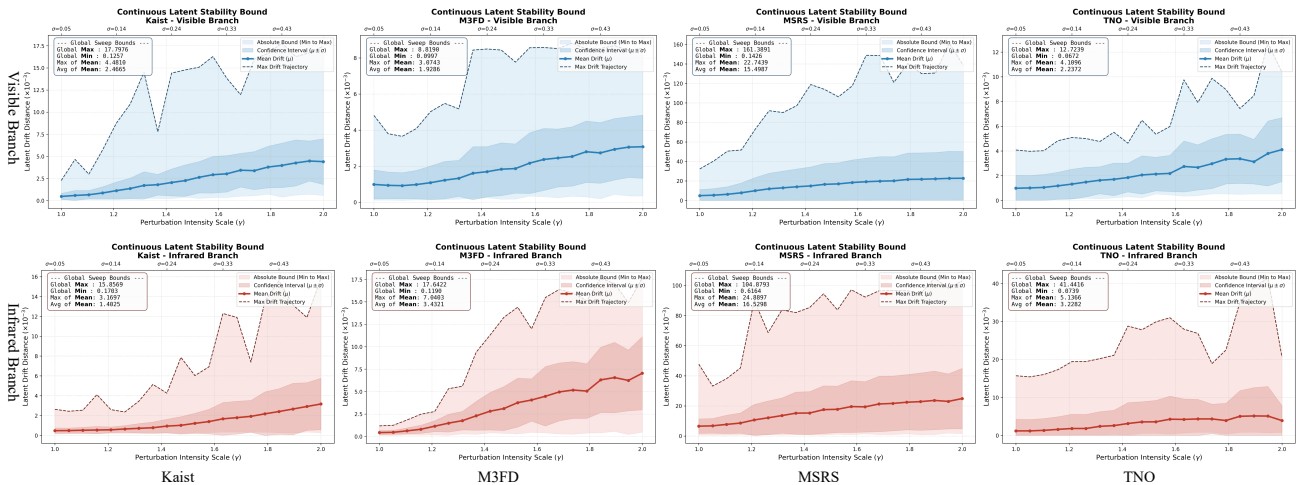

*Figure 15.* **Trajectory Visualizations of the Continuous Parameter Sweep.** MSE of the latent feature drift as perturbation intensity scales incrementally ($\gamma \in [1.0, 2.0]$ and $\sigma \in [0.05, 0.5]$) across Kaist, M3FD, MSRS, and TNO (Top: Visible $\mathcal{V}$, Bottom: Infrared $\mathcal{I}$). The strictly bounded growth indicates the structural stability of the content anchors under continuous manifold shifts.

*Table 6.* **Quantitative Assessment of Latent Stability under Stochastic Perturbation.** We report the statistics (Mean ($\mu$), Std ($\sigma$), Max) of the feature drift vector ($\times 10^{-3}$) after subjecting the attribute encoder to extreme manifold shifts. The negligible drift values across all datasets suggest that the content anchors remain structurally stable despite significant attribute variations.

| Dataset | Modality | Latent Drift Statistics ($\times 10^{-3}$) | | |
| --- | --- | --- | --- | --- |
| | | Mean ($\mu$) ↓ | Std ($\sigma$) ↓ | Max ↓ |
| Kaist | Visible | 2.4905 | 1.7075 | 13.6590 |
| | Infrared | 1.0117 | 1.0892 | 9.0494 |
| M3FD | Visible | 1.8907 | 1.4526 | 7.9797 |
| | Infrared | 3.2394 | 1.9399 | 10.1430 |
| MSRS | Visible | 17.0580 | 20.6720 | 112.2000 |
| | Infrared | 17.8050 | 14.9700 | 78.5620 |
| TNO | Visible | 2.2502 | 1.6363 | 7.5547 |
| | Infrared | 3.3138 | 3.2175 | 17.0900 |

**Stability Analysis of Content Anchors.** To establish a baseline of robustness under considerable but fixed perturbation, we initially scale the attribute magnitude by a factor of $\gamma = 1.5$ and inject Gaussian noise ($\sigma = 0.2$). We then quantify the structural invariance of the latent space by calculating the feature drift metric, defined as the $L_2$ distance between the original content code and the re-encoded feature derived from this attribute-shifted synthesis. As visualized in Figure 14, the error distributions across all test datasets are heavily skewed towards zero, indicating high stability despite the substantial stochastic variations applied to the attribute channel. Specifically, as depicted in Table 6, the mean drift is maintained at a negligible level, exemplified by the infrared branch of the Kaist dataset at approximately 0.0010. This robustness is consistent across

statistical metrics, with the standard deviation on the M3FD visible branch restricted to as low as 0.0015. Furthermore, the system demonstrates high resilience against extreme outliers. Although the MSRS dataset exhibits a slightly elevated response compared to other datasets, the maximum observed drift across all test sets remains strictly bounded within a low-magnitude range ($< 0.12$). These quantitative results demonstrate that the LaRA-Fusion encoder has learned to effectively decouple geometric content from attribute variations. By maintaining such bounded structural consistency even under intense manifold shifts, the model indicates its capability to mitigate numerical shortcuts, suggesting that the fusion process relies on stable semantic representations.

**Continuous Parameter Sweep for Sensitivity Analysis.** To assess whether structural stability derives from intrinsic architecture rather than specific hyperparameter configurations, we conduct a continuous parameter sweep across all testing datasets. By systematically escalating the perturbation intensity over twenty incremental steps, we scale the attribute magnitude ($\gamma \in [1.0, 2.0]$) while simultaneously injecting increasing levels of Gaussian noise ($\sigma \in [0.05, 0.5]$) and recording the MSE of the latent feature drift. As illustrated in Figure 15, the resulting trajectories indicate that the average latent drift exhibits a progressive yet strictly bounded growth as perturbation increases. Notably, even on the highly responsive MSRS dataset, the maximum drift trajectory remains tightly constrained within a moderate bound without triggering feature divergence or geometric collapse. These findings suggest that the disentangled content anchors ($\boldsymbol{C}$) maintain robustness across a broad range of manifold shifts, supporting the model's high tolerance to diverse stochastic sampling conditions.

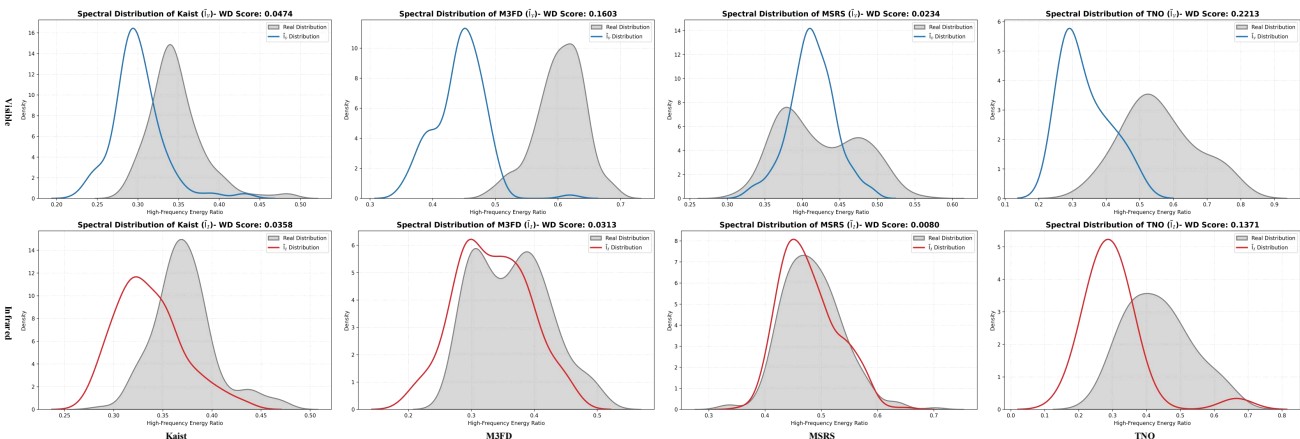

*Figure 16.* **Global Spectral Fidelity Analysis.** KDE plots comparing high-frequency energy distributions between real and cycle-reconstructed images across Kaist, M3FD, MSRS, and TNO (Top: Visible $\mathcal{V}$, Bottom: Infrared $\mathcal{I}$). The distributions demonstrate high statistical overlap with low Wasserstein distance (WD) scores.

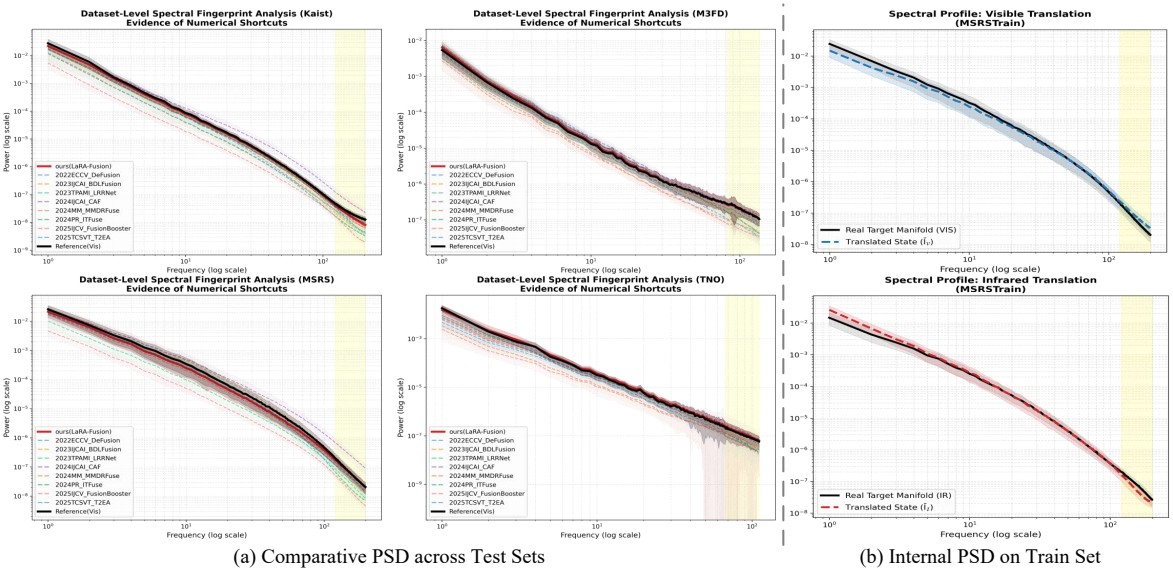

(a) Comparative PSD across Test Sets      (b) Internal PSD on Train Set

*Figure 17.* **1D PSD Analysis.** (a) illustrates the comparative global average PSD across test datasets, indicating that LaRA-Fusion mitigates the high-frequency uplifts or sags observed in several evaluated baselines. (b) details the internal PSD of intermediate translations on the training set for both source modalities, which closely follows the intrinsic $1/f$ frequency decay.

## G. Spectral Fidelity and Statistical Naturalness

Beyond spatial robustness, we further evaluate the spectral integrity of the generated images to ensure the encoder relies on semantic structures rather than high-frequency artifacts. We compute the high-frequency energy ratio for both real and cycle-reconstructed images across the test sets. As shown in Figure 16, the generated distributions closely track the real data statistics. The Wasserstein distance (WD) scores remain low, recording 0.0080 for the MSRS infrared branch and 0.0474 for the Kaist visible branch. While legacy datasets like TNO exhibit slightly higher deviations due to intrinsic sensor noise, the overall distributions display a consistent slight leftward shift. This phenomenon indicates that LaRA-Fusion functions as a conservative generator that suppresses inherent sensor noise while mitigating the spectral uplift typically associated with degenerate shortcut learning. The absence of anomalous high-frequency tails suggests that the model synthesizes natural representations without severe artifacts.

Consequently, a detailed 1D Power Spectral Density (PSD) (Kay & Marple, 2005) analysis provides further direct assessment regarding the presence of these degenerate shortcuts. As illustrated in Figure 17(b), the internal PSD curves of our intermediate translations closely approximate the natural $1/f$ decay without severe high-frequency anomalies.

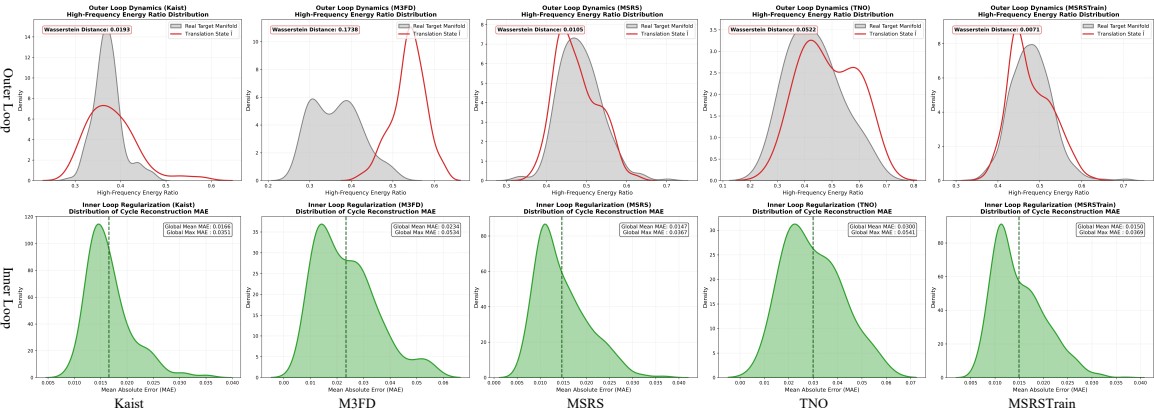

*Figure 18.* **Internal State Analysis of Dual-Loop Constraints.** The visualizations illustrate the internal state dynamics across different datasets. **Outer Loop:** KDE of high-frequency energy ratios, demonstrating that intermediate translation states align closely with the real target distributions, characterized by low Wasserstein distances. **Inner Loop:** Density distributions of the cross-modal cycle reconstruction MAE, which remain consistently concentrated near zero, reflecting stable geometric invertibility.

Furthermore, comparative evaluations across multiple test sets (Figure 17(a)) reveal that while several baselines exhibit high-frequency uplifts (noise injection) or structural sags (texture collapse), LaRA-Fusion maintains a closer distribution matching with the target data.

## H. Pixel-level Fidelity and Reconstruction Stability

Having evaluated the theoretical robustness and spectral naturalness of the proposed framework, we conclude the quantitative analysis by examining pixel-level fidelity. We utilize boxplots to visualize the error distribution for both self-reconstruction and cycle-reconstruction tasks across four diverse testing datasets. As illustrated in Figure 19, the error metrics exhibit consistently low median values and compact interquartile ranges, indicating that the model achieves reliable restoration across varied scenarios. This statistical stability implies that the dual-loop mechanism preserves intrinsic semantic information during complex cross-modal translations, further supporting that the final fusion results are grounded in authentic content rather than hallucinated artifacts.

## I. Internal State Analysis of Dual-Loop Constraints

Building upon the pixel-level stability demonstrated previously, we further investigate the internal state dynamics of the proposed framework to understand the necessity and individual regularization effects of the dual-loop constraints. A dataset-level analysis suggests that both loops play integral, complementary roles without either dominating the optimization process. Specifically, the inner loop anchors the geometric topology, as indicated by the cross-modal

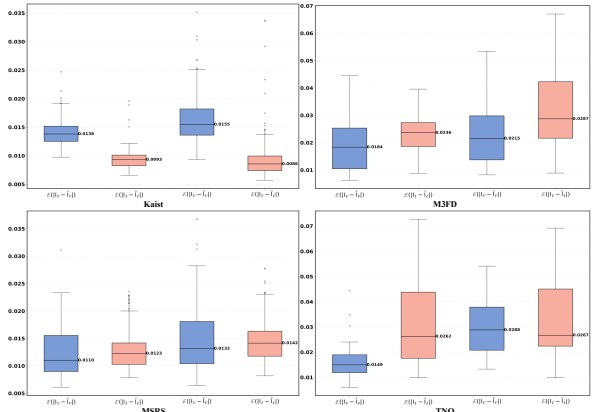

*Figure 19.* **Distribution of reconstruction fidelity across testing datasets** (Top-Left: Kaist, Top-Right: M3FD, Bottom-Left: MSRS, Bottom-Right: TNO). The boxplots quantify the MAE for both self-reconstruction ($\mathbb{E}[\|\mathbf{I} - \hat{\mathbf{I}}\|]$) and cycle-reconstruction ($\mathbb{E}[\|\mathbf{I} - \tilde{\mathbf{I}}\|]$). The consistently low medians and narrow interquartile ranges indicate that LaRA-Fusion maintains reliable pixel-level fidelity and stability, with minimal information loss occurring during the cross-modal translation loops.

cycle reconstruction remaining tightly concentrated near zero across varied scenes. Concurrently, the outer loop regulates the statistical distribution of the latent space. As illustrated by the high-frequency energy ratios in Figure 18, the intermediate translation states align closely with authentic target distributions, yielding low Wasserstein distances. This adversarial matching mechanism discourages the network from embedding degenerate high-frequency noise merely to satisfy cycle-consistency objectives. Consequently, the loops function synergistically: the inner loop preserves structural invertibility while the outer loop encourages distribution matching, jointly mitigating latent space collapse.

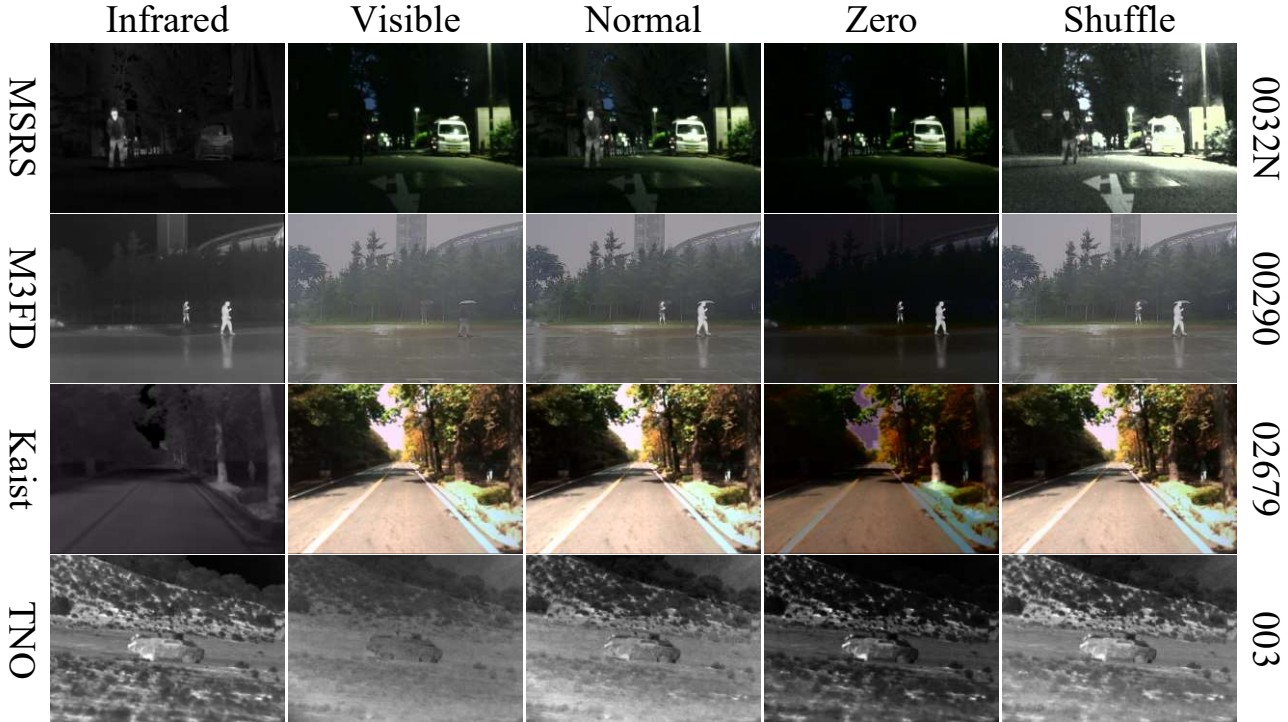

*Figure 20.* **Visual assessment of functional separation.** Qualitative results obtained by either zeroing the attribute vectors or injecting randomly shuffled attributes. The retention of base spatial outlines under arbitrary attribute assignments indicates that structural geometry is preserved independently of the modality-specific state.

*Table 7.* **Quantitative Analysis of Functional Independence.** Performance metrics evaluated across four datasets when the attribute branch is subjected to zeroing or random shuffling, with the best and second-best values in **bold** and underline.

| Dataset | Mode | EN | SD | AG | MI | VIF | $Q_{abf}$ |
|---------|------|-----|------|------|------|------|------|
| **Kaist** | Normal | **7.14** | 59.67 | 3.58 | **3.53** | **0.97** | **0.70** |
| | Zero | 6.53 | 38.48 | 2.25 | 2.19 | 0.76 | 0.36 |
| | Shuffle | 7.07 | **61.51** | **3.78** | 3.39 | **0.97** | 0.62 |
| **M3FD** | Normal | **7.02** | **41.67** | **5.33** | **3.28** | 0.86 | **0.65** |
| | Zero | 6.20 | 30.91 | 2.49 | 2.14 | 0.53 | 0.24 |
| | Shuffle | 6.98 | 41.40 | 5.15 | **3.28** | **0.87** | 0.56 |
| **MSRS** | Normal | **6.77** | **45.73** | **3.76** | **2.98** | 1.03 | **0.66** |
| | Zero | 4.79 | 20.15 | 1.32 | 1.69 | 0.51 | 0.22 |
| | Shuffle | 6.64 | 44.42 | 3.47 | 2.82 | **1.06** | 0.43 |
| **TNO** | Normal | **7.10** | 50.95 | **4.90** | **2.52** | **0.82** | **0.49** |
| | Zero | 6.49 | 33.61 | 3.05 | 1.64 | 0.56 | 0.31 |
| | Shuffle | 7.04 | **51.50** | 4.76 | 2.43 | 0.75 | 0.40 |

## J. Mechanism and Functional Independence of the Attribute Branch

Having established the synergistic regularization of the dual-loop constraints on the global latent space, we now focus on the internal mechanisms of the attribute branch. While the dual-loop framework supports overall topological con-sistency, understanding LaRA-Fusion requires examining how transient environmental characteristics are navigated and subsequently isolated from the geometric structures.

The framework navigates the attribute manifold using a dynamic gating mechanism, where the stochastic factor $\alpha$ is injected as a prior bias into the logit domain to simulate continuous environmental shifts. This stochastic traversal acts as a regularization strategy, guiding the content encoder to maintain stable semantic representations regardless of the imposed attribute state. To determine whether the structural geometry is decoupled from such transient variations, we further evaluate the network's response to extreme attribute manipulations. Specifically, replacing the extracted attributes with zero tensors leads to a decline in quantitative performance (Table 7) and produces flat structural outlines devoid of thermal or illumination saliency (Figure 20), indicating that the attribute branch governs modality appearance. Conversely, injecting randomly shuffled attributes keeps the spatial geometry visually stable, suggesting that the content anchor independently retains geometric structure. However, this mismatch degrades semantic coherence and quantitative metrics (e.g., Kaist $Q_{abf}$: 0.70 → 0.62). Ultimately, these observations suggest that preserving the originally matched attribute correspondences yields better semantic fidelity and overall performance.

*Table 8.* **Quantitative comparison of computational efficiency.** The evaluation includes the number of parameters (M), FLOPs (G), and average inference time (s) per image, assessed on an NVIDIA A800 GPU using standard $256 \times 256$ inputs.

| Method | Source | Params(M)↓ | FLOPs(G)↓ |
|---|---|---|---|
| DeFusion | ECCV'22 | 7.8745 | 38.6390 |
| BDLFusion | IJCAI'23 | 32.7238 | 80.9720 |
| LRRNet | TPAMI'23 | 0.0492 | 6.0445 |
| CAF | IJCAI'24 | 0.1483 | 19.4532 |
| MMDRFuse | ACMMM'24 | **0.0001** | **0.0142** |
| ITFuse | PR'24 | 0.0825 | 11.3514 |
| FusionBooster | IJCV'25 | 0.5977 | 29.6726 |
| LUTFuse | ICCV'25 | 0.0078 | 1.0339 |
| T2EA | TCSVT'25 | 0.2754 | 104.7457 |
| LaRA-Fusion | This Paper | 4.0820 | 53.5504 |

## K. Computational Efficiency Analysis

To evaluate practical deployment viability, we conduct a computational overhead analysis measuring total parameters and FLOPs across the comparative methods. For consistency, this evaluation operates on $256 \times 256$ random tensors using an NVIDIA A800 GPU. To ensure measurement stability, we record the average computational overhead over ten independent trials of one hundred iterations each, following a standard fifty-iteration hardware warm-up phase managed by high-precision CUDA events.

The corresponding results are detailed in Table 8. Compared to other evaluated methods, LaRA-Fusion maintains a moderate profile in terms of parameter count and computational demands. We acknowledge that certain lightweight architectures achieve fewer parameters and lower FLOPs through their specialized designs. For instance, MMDRFuse utilizes knowledge distillation, and LUTFuse distills features into learnable look-up tables. Consequently, LaRA-Fusion presents a trade-off, balancing moderate computational overhead with fusion quality and topological robustness.

## L. Downstream Applications

Following the computational efficiency analysis, we further explore the effectiveness of LaRA-Fusion by extending our evaluation to typical downstream semantic tasks, a common practice in related low-level vision tasks (Xu et al., 2024b; 2025d; Niu et al., 2025; Huang et al., 2025a; Li et al., 2025).

**Object Detection Setup.** We randomly sample 800 image pairs from the M3FD dataset and divide them into training, validation, and test sets with an 8:1:1 ratio to perform object detection. We employ YOLOv5 (Khanam & Hussain, 2024) to evaluate the detection performance using the mAP@50 metric. The network is optimized via the SGD optimizer with an initial learning rate of 0.01 and a batch size of 64, trained over 100 epochs.

*Table 9.* Quantitative comparison on object detection task, with the best and second-best values in **bold** and underline.

| Method | Source | M3FD Object Detection | | | | | | |
|---|---|---|---|---|---|---|---|---|
| | | mAP50 | People | Car | Bus | Lamp | Motor | Truck |
| DeFusion | ECCV'22 | 65.99 | 74.69 | 86.61 | 84.94 | 55.61 | 38.95 | 55.18 |
| BDLFusion | IJCAI'23 | 64.60 | 73.86 | 86.17 | 80.92 | 57.36 | 40.89 | 48.37 |
| LRRNet | TPAMI'23 | 65.99 | 72.26 | 86.10 | 85.68 | 56.64 | 45.96 | 49.29 |
| CAF | IJCAI'24 | 64.15 | 74.30 | 86.63 | 80.60 | 56.36 | 37.72 | 49.26 |
| MMDRFuse | ACMMM'24 | 66.04 | 74.01 | 86.42 | 87.45 | 57.21 | 37.72 | 53.42 |
| ITFuse | PR'24 | 63.12 | 73.23 | 86.09 | 77.07 | 52.11 | 38.62 | 51.58 |
| FusionBooster | IJCV'25 | 66.55 | **74.77** | 86.37 | **89.61** | 55.75 | 40.59 | 52.19 |
| LUTFuse | ICCV'25 | 64.71 | 70.85 | 85.23 | 79.68 | 56.34 | 38.98 | **57.18** |
| T2EA | TCSVT'25 | 65.36 | 73.56 | 86.54 | 85.76 | 55.62 | 40.30 | 50.37 |
| **LaRA-Fusion** | This Paper | **66.74** | 69.62 | **87.80** | 87.15 | **58.04** | **48.27** | 49.59 |

*Table 10.* Quantitative comparison on semantic segmentation task, with the best and second-best values in **bold** and underline.

| Method | Source | MSRS Semantic Segmentation | | | | | |
|---|---|---|---|---|---|---|---|
| | | mIoU | background | car | person | bike | car stop |
| DeFusion | ECCV'22 | 64.96 | 96.93 | 82.91 | 56.12 | 60.12 | 28.74 |
| BDLFusion | IJCAI'23 | 64.69 | 96.91 | 82.48 | 56.55 | 59.08 | 28.44 |
| LRRNet | TPAMI'23 | 65.09 | 96.91 | 82.71 | 55.40 | 59.10 | **31.31** |
| CAF | IJCAI'24 | 58.61 | 96.44 | 75.39 | 55.99 | 55.80 | 9.43 |
| MMDRFuse | ACMMM'24 | 65.50 | 96.96 | 82.46 | 56.93 | 60.31 | 30.86 |
| ITFuse | PR'24 | 64.39 | 96.92 | **82.93** | 56.17 | 59.61 | 26.34 |
| FusionBooster | IJCV'25 | 64.49 | 96.92 | 82.82 | 55.12 | 59.36 | 28.22 |
| LUTFuse | ICCV'25 | **65.60** | **96.97** | 82.76 | **57.06** | 59.96 | 31.23 |
| T2EA | TCSVT'25 | 65.20 | 96.94 | 82.35 | 55.85 | 60.60 | 30.25 |
| **LaRA-Fusion** | This Paper | 65.53 | 96.96 | 82.58 | 55.62 | **61.37** | 31.10 |

**Semantic Segmentation Setup.** We perform semantic segmentation on the MSRS dataset using DeepLabV3+ (Chen et al., 2018) as the backbone to evaluate the mIoU. To mitigate long-tail imbalance, we exclude minority categories that yield zero accuracy across the comparative methods, thereby computing an adjusted mIoU. The training utilizes the SGD optimizer with a batch size of 32 and an initial learning rate of $7 \times 10^{-3}$ for 100 epochs, where the backbone is frozen for the first 50 epochs.

**Comprehensive Analysis.** For object detection, as detailed in Table 9 and Figure 21, LaRA-Fusion achieves the highest mAP@50 (66.74%), indicating its capability to provide effective representations for identifying diverse objects. For semantic segmentation, Table 10 and Figure 22 show that our method maintains a competitive mIoU (65.53%), trailing the top-performing method by 0.07%. These results collectively suggest that the fused representations generated by LaRA-Fusion provide practical utility for downstream perception applications.

## M. Potential Limitations

While LaRA-Fusion demonstrates robust performance, certain limitations remain. If one sensor considerably degrades (e.g., suboptimal infrared quality in specific Kaist samples, as illustrated in Figure 20), the fused outcome may naturally favor the more informative modality. Methodologically, although patch-based training stabilizes optimization by filtering uninformative regions, it inherently restricts the cap-

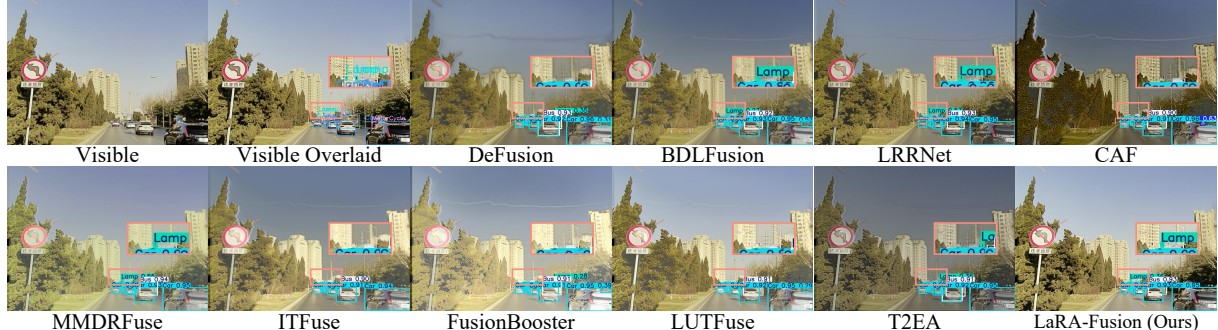

Figure 21. Qualitative comparison of LaRA-Fusion with nine state-of-the-art methods on the object detection task.

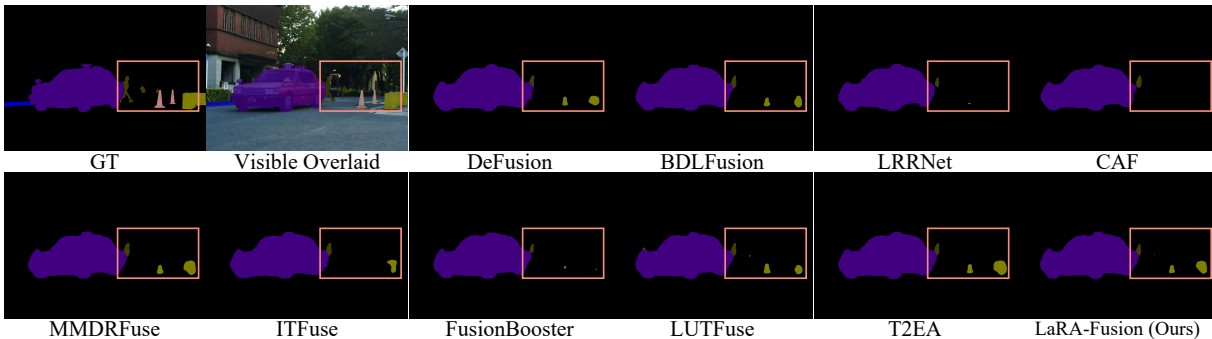

Figure 22. Qualitative comparison of LaRA-Fusion with nine state-of-the-art methods on the semantic segmentation task.

ture of broader global contexts, highlighting a direction for future full-image training paradigms. Architecturally, the disentangled dual-stream design couples specialized modules to specific routing paths, complicating the adoption of off-the-shelf monolithic backbones. Nevertheless, the internal feature extraction blocks are flexible and can be substituted with alternatives containing more or fewer parameters. Finally, the overall parameter count and high FLOPs suggest that deploying LaRA-Fusion in strictly real-time or resource-constrained edge environments may present practical challenges.

## N. Additional Visual Comparisons

Finally, an additional visual comparison is provided to further illustrate reconstruction-cycle fidelity and qualitative fusion results against advanced methods across different datasets.

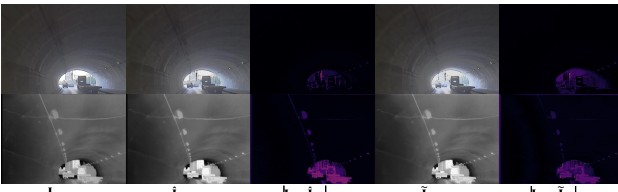

Figure 24. Visual reconstruction examples from M3FD dataset.

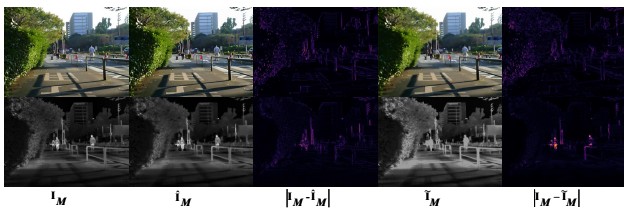

Figure 25. Visual reconstruction examples from MSRS dataset.

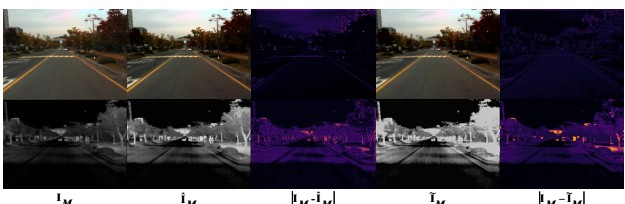

Figure 23. Visual reconstruction examples from Kaist dataset.

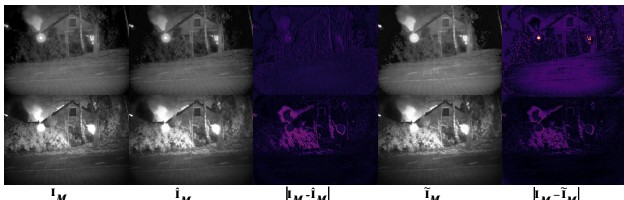

Figure 26. Visual reconstruction examples from TNO dataset.

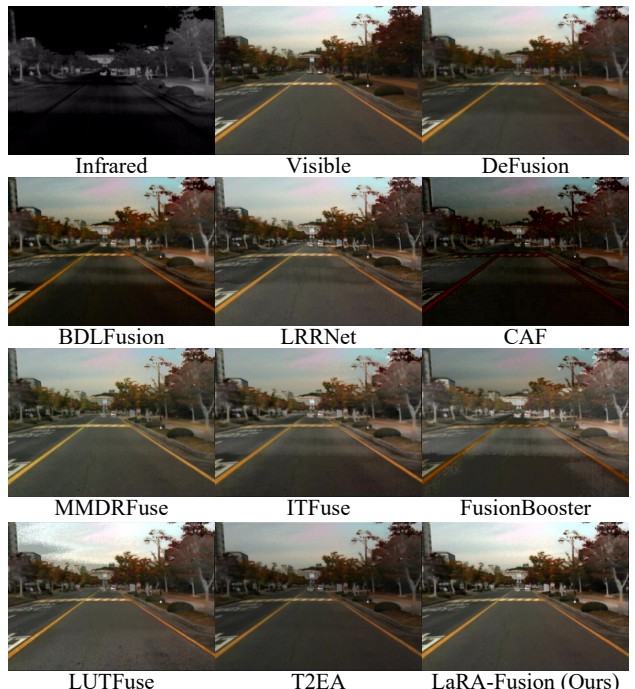

Figure 27. More visual comparisons against state-of-the-art methods from the Kaist dataset.

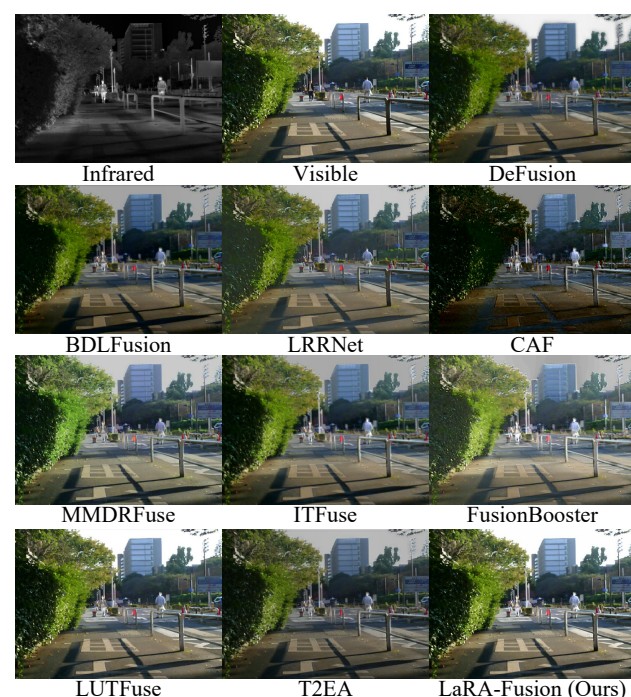

Figure 29. More visual comparisons against state-of-the-art methods from the MSRS dataset.

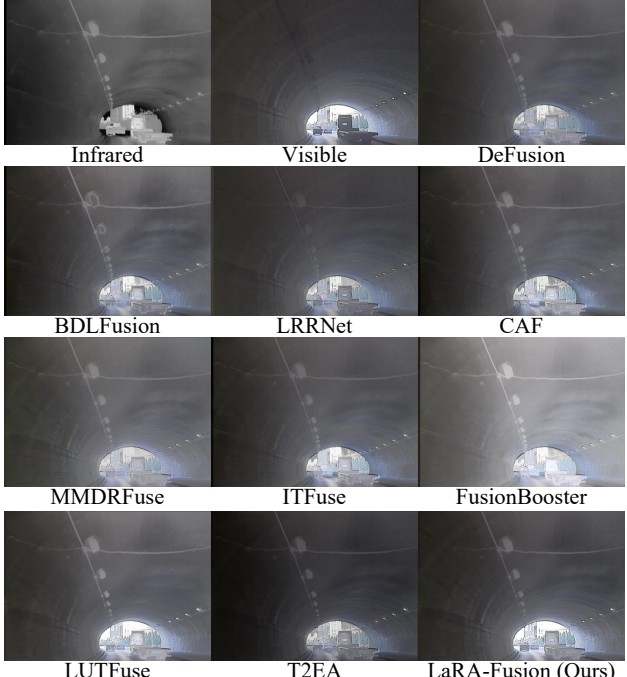

Figure 28. More visual comparisons against state-of-the-art methods from the M3FD dataset.

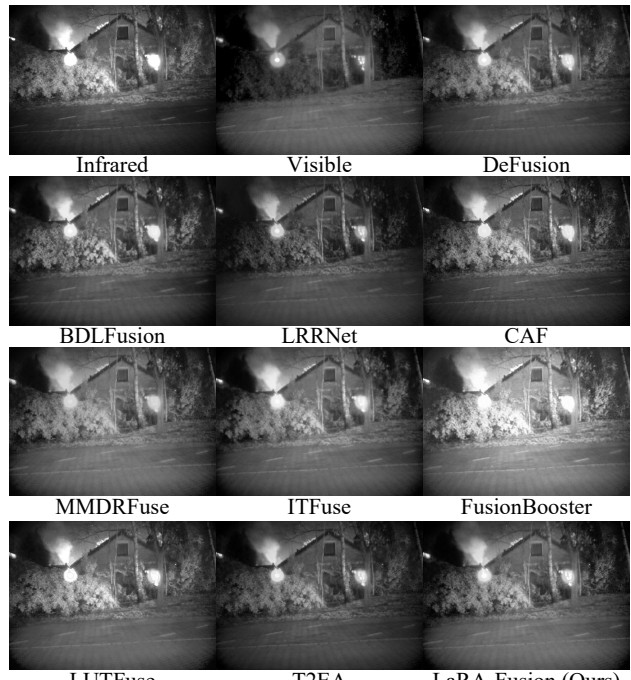

Figure 30. More visual comparisons against state-of-the-art methods from the TNO dataset.

