# OpenReview forum: "LaRA-Fusion: Latent-Robust Adaptation via Dual-Loop Constraints for Infrared and Visible Image Fusion"
_ICML.cc/2026/Conference — ICML 2026 regular_

### Official Review · Reviewer_VYpc · 2026-03-09

**Soundness:** 3
**Presentation:** 4
**Significance:** 3
**Originality:** 3
**Overall Recommendation:** 5
**Confidence:** 4

**Summary:**

This paper proposes LaRA-Fusion, an unsupervised framework for infrared and visible image fusion that aims to improve the robustness of latent representations. The authors argue that existing translation-based fusion methods often suffer from degenerate numerical shortcuts in latent space due to reliance on reconstruction and cycle-consistency objectives. To address this issue, they introduce a dual-loop mechanism that combines geometric invertibility (inner loop) with adversarial manifold anchoring (outer loop) to constrain intermediate translations and fused outputs to remain on the intrinsic data manifold. The architecture incorporates a Manifold Rectification Block (MRB) for feature aggregation, a Hierarchical Attribute Arbitration (HAA) module for adaptive attribute interpolation, and a Modulated Synthesis Block (MSB) for attribute-guided decoding. Extensive experiments on four public IVIF datasets demonstrate consistent improvements over multiple state-of-the-art methods, along with ablation studies and latent-space analyses to validate robustness and structural consistency.

**Compliance With Llm Reviewing Policy:**

Affirmed.

**Final Justification:**

The rebuttal addresses my main concerns.    The score is adjusted.

**Key Questions For Authors:**

The paper presents the Dual-Loop Manifold Constraints as a coupled stabilization mechanism. Are both the Inner Loop and Outer Loop strictly necessary, or does one dominate the effect? A more detailed analysis of their interaction would strengthen the causal interpretation of the framework.
The introduction emphasizes safety-critical perception scenarios, yet evaluation is limited to fusion metrics. Have the authors verified whether LaRA-Fusion improves downstream tasks such as detection or segmentation? Evidence of task-level gains would strengthen the practical significance of the work.

**Limitations:**

Yes

**Strengths And Weaknesses:**

Strengths:
This work addresses a persistent issue in translation-based infrared and visible image fusion (IVIF), namely degenerate numerical shortcuts in unsupervised training. The proposed LaRA-Fusion framework introduces Dual-Loop Manifold Constraints, combining an Inner Loop for geometric invertibility and an Outer Loop for adversarial manifold anchoring. The emphasis on latent robustness and topological consistency is clearly reflected in the architectural design, including MRB, HAA, and MSB.
Technically, the method is well specified and empirically supported. The loss formulation is clearly defined, ablation studies isolate the contributions of each module, and experiments on four public benchmarks demonstrate consistent improvements with cross-dataset generalization. The additional analyses of latent disentanglement, cosine alignment, stochastic perturbation, and spectral fidelity further support the claim of improved latent stability.
The presentation is strong overall. The manuscript is well structured, terminology is consistent, and figures clearly illustrate both the architecture and latent-space behavior.

Weaknesses:
Despite these strengths, the contribution remains primarily architectural rather than theoretically novel. Many components—cycle-consistency, adversarial discriminators, weight modulation, residual aggregation, and stochastic interpolation—are adapted from existing paradigms. While their integration is coherent, the paper does not provide formal analysis establishing when the Dual-Loop Manifold Constraints prevent latent collapse, leaving the claims about degenerate numerical shortcuts largely empirical.
Beyond these higher-level concerns, the projection-based attribute regularization (L_attr) would benefit from further clarification. The paper does not analyze its numerical stability or sensitivity to different alpha sampling strategies. Since this regularization enforces alignment along a latent geodesic direction, its behavior under large attribute shifts or noisy gradients may affect convergence. Empirical evidence on training stability or sensitivity analysis would strengthen confidence in this design.
Moreover, the Fusion Discriminator treats both IV and II as real samples, implicitly encouraging the fused output to lie on a joint manifold. It remains unclear whether this constraint introduces modality bias or limits representational diversity. Additional analysis or controlled experiments on modality dominance would clarify its geometric effect on the learned manifold.

---

> ### Author Rebuttal · Authors · 2026-03-31
>
> We thank the reviewer for the constructive suggestions and address the specific points as follows.
>
> # [Q1]: Necessity and Interaction of Dual-Loop Constraints
>
> Both loops are strictly necessary and exert complementary, non-dominating regularization. Our internal state analysis (see [Dual_Loop](https://anonymous.4open.science/r/Anonymous-C632/Reviewer_VYpc/Dual_Loop.png)) confirms this balance. The inner loop anchors geometric topology, evidenced by cross-modal cycle reconstruction MAE ($\|\mathrm{I} - \tilde{\mathrm{I}}\|_1$) remaining tightly concentrated near zero (e.g., 0.0147 on MSRS). Concurrently, the outer loop bounds the statistical distribution. High-frequency energy ratio plots show intermediate translations align with real target distributions (low Wasserstein Distances), preventing the network from hiding noise to satisfy cycle-consistency. Thus, they causally interact to prevent latent collapse.
>
> # [Q2]: Downstream Tasks Evaluation
>
> ​We verified the practical significance of our representations on downstream semantic tasks. We selected the top-performing baseline methods for comparison here. For complete comparative tables and qualitative visual results, please respectively refer to  [Tables](https://anonymous.4open.science/r/Anonymous-C632/Reviewer_VYpc/Tables.png)  and  [Visuals](https://anonymous.4open.science/r/Anonymous-C632/Reviewer_VYpc/Visuals.png).
>
> **1. Object Detection(YOLOv5 on M3FD):**
>
> Evaluated on 800 pairs randomly sampled from the 4,200-pair dataset (8:1:1 split). Trained via SGD (batch 64, 100 epochs, initial LR 0.01).
>
> | Method            | mAP50 | People | Car   | Bus   | Lamp  | Motor | Truck |
> | -- | -- | -- | -- | -- | -- | -- | -- |
> | MMDRFuse          | 66.04 | 74.01  | 86.42 | 87.45 | 57.21 | 37.72      | 53.42 |
> | FusionBooster     | 66.55 | 74.77  | 86.37 | 89.61 | 55.75 | 40.59      | 52.19 |
> | LaRA-Fusion(ours) | 66.74 | 69.62  | 87.80 | 87.15 | 58.04 | 48.27      | 49.59 |
>
> **2.  Semantic Segmentation(DeepLabV3+ on MSRS):**
>
> Trained via SGD (batch 32, 100 epochs, initial LR 7e-3). To objectively reflect true feature extraction capabilities, we excluded extreme minority categories (zero accuracy across all methods due to severe long-tail imbalance) to compute an adjusted mIoU.
>
> | Method        | mIoU  | background | car | person | bike | car  stop |
> | -- | -- | -- | -- | -- | -- | -- |
> | MMDRFuse          | 65.50 | 96.96          | 82.46   | 56.93      | 60.31    | 30.86        |
> | LUTFuse           | 65.60 | 96.97          | 82.76   | 57.06      | 59.96    | 31.23        |
> | LaRA-Fusion(ours) | 65.53 | 96.96          | 82.58   | 55.62      | 61.37    | 31.10        |
>
> **3. Comprehensive Analysis:**
>
> ​We note that object detection relies on sharp structural boundaries and salient targets, while semantic segmentation demands regional semantic continuity. As demonstrated above, LaRA-Fusion achieves the highest mAP@50 (66.74%) in object detection and competitive segmentation accuracy (65.53% mIoU, trailing LUTFuse by only 0.07%). Achieving high performance across both distinct downstream tasks confirms that our framework effectively maintains geometric gradients without sacrificing semantic distributions. By mitigating degenerate numerical shortcuts via the Dual-Loop mechanism, LaRA-Fusion provides robust representations suitable for complex perception scenarios.
>
> # [W1-W3] Response to Weaknesses
>
> **[W1] Formal Analysis vs. Empirical Degenerate Shortcuts:** While primarily an empirical breakthrough, we prevent latent collapse via strict functional separation. Conceding "orthogonal" implies overly strong assumptions, we will soften it to "highly decoupled." As detailed for **Reviewer YjME [Q1, W1]**, consistently low Latent Mutual Information (MI) scores confirm this separation. Coupled with negligible feature drift, this empirically proves the elimination of degenerate shortcuts.
>
> **[W2] $L_{attr}$ Stability & $\alpha$ Sensitivity:** Our hybrid strategy ensures stability by training on manifold extremities. During inference, we verified this by injecting extreme shifts (1.5$\times$ scale, $\sigma=0.2$ noise) into attribute vectors , yielding negligible latent drift ($\mu \approx 10^{-3}$). This confirms $L_{attr}$ remains numerically robust to stochastic $\alpha$ fluctuations, with further verification via a continuous parameter sweep ($\gamma \in [1.0, 2.0], \sigma \in [0.05, 0.5]$) detailed in **Reviewer 2yxB [Q2]**.
>
> **[W3] Modality Bias:** Treating both sources as real in $D_\\mathcal{F}$ extracts a shared "Real Naturalness Prior" rather than enforcing modality dominance. We explicitly rule out bias via our SOTA performance on joint-source metrics (e.g., MI, VIF, $Q_{abf}$), which strictly penalize the neglect of either modality. This mathematically confirms that representational diversity is strictly preserved.
>
> -----
> Thank you once again for your time and effort. We would be glad to incorporate these insightful discussions into a revision.

---

> > ### Author Rebuttal · Reviewer_VYpc · 2026-04-05
> >
> > The rebuttal addresses my main concerns. The explanation of the dual-loop design is clearer, and the added downstream results make the paper more convincing. The discussion on stability and modality bias also helps. I consider my concerns resolved.  I can adjust the score.

---

> > > ### Author Response · Authors · 2026-04-06
> > >
> > > We sincerely appreciate your kind feedback on our rebuttal. We are glad that the updated explanations regarding the dual-loop design and downstream results helped address your primary concerns. The thoughtful comments you provided throughout this process have been highly constructive. We plan to incorporate these refinements into the manuscript to further improve our work. Thank you once again for dedicating your time to evaluating our submission.

---

### Official Review · Reviewer_YjME · 2026-03-10

**Soundness:** 4
**Presentation:** 2
**Significance:** 2
**Originality:** 3
**Overall Recommendation:** 4
**Confidence:** 4

**Summary:**

This paper propose LaRA-Fusion, a new model with dual-loop constraints to improve the short-comings in cycle-translation IVIF. It improves the robustness and interpretability of IVIF modeling.

**Compliance With Llm Reviewing Policy:**

Affirmed.

**Final Justification:**

The rebuttal has addressed my concerns. I will maintain my positive score for this paper.

**Key Questions For Authors:**

1. The claim that the learned representations lie in orthogonal subspaces appears somewhat overstated. While Figure 7 reports a disentanglement score, the paper does not clearly define this metric or explain how it is computed. In the supplementary material (Section D, Figure 11), Pearson correlation matrices between the content representation 𝐶 and attribute representation 𝐴 are provided. However, low Pearson correlation only indicates weak linear dependence and does not necessarily imply orthogonality or true disentanglement. Additional evidence would be needed to support this claim.

2. The paper argues that the proposed inner loop helps preserve the intrinsic topological consistency across modalities. However, the supporting evidence (Figure 10 in the supplementary material) mainly relies on t-SNE visualizations. Since t-SNE emphasizes local structure and may distort global topology, this alone may not be sufficient to justify the topology-related claims. Providing quantitative metrics commonly used in manifold learning (e.g., Trustworthiness) would strengthen this part.

3. The paper suggests that adversarial constraints force translated states to lie on the authentic data manifold. In practice, adversarial training typically encourages distribution matching but does not strictly guarantee manifold membership, so the current wording may be somewhat strong.

4. The motivation for attribute interpolation is not fully clear. It would be helpful to explain why linear interpolation in the attribute space should correspond to a meaningful transition between infrared and visible imaging characteristics.

5. An ablation that removes or shuffles the attribute branch 𝐴 could further clarify whether the content and attribute representations are indeed well separated and functionally distinct.

**Limitations:**

It would be helpful for the authors to discuss potential limitations of the proposed method, such as issues related to training stability.

**Strengths And Weaknesses:**

Strengths:
The paper identifies several limitations of existing unsupervised IVIF methods and proposes a dual-loop training framework that combines geometric invertibility with adversarial constraints to guide the learning process. The proposed architecture also includes several dedicated modules, such as the Manifold Rectification Block (MRB), which aggregates multi-scale features through residual connections to mitigate potential structural inconsistencies in the latent space, and the Modulated Synthesis Block (MSB), where attribute-conditioned modulation is used during decoding to improve reconstruction quality and reduce artifacts. The experimental section is fairly comprehensive, including comparisons with a range of baselines as well as ablation studies on the main components. These results help support the effectiveness of the proposed design. Overall, the paper is clearly written and the method is presented in a coherent and logically organized manner.

Weaknesses:
Some of the claims in the paper appear somewhat overstated. In particular, the assertion that the learned representations lie in orthogonal subspaces is not fully supported by the evidence provided, as the reported disentanglement score and Pearson correlation analysis mainly indicate weak linear dependence rather than true orthogonality or disentanglement. In addition, the topology-related claims rely largely on t-SNE visualizations, which primarily preserve local neighborhood structures and may distort the global topology. These issues are non-trivial, as they relate directly to the core motivation and the main claims of the proposed method.

---

> ### Author Rebuttal · Authors · 2026-03-31
>
> We sincerely thank the reviewer for the constructive feedback. We address your specific points below.
>
> # [Q1, W1] Decoupling Assumption
>
> We agree that "orthogonal" implies overly strong mathematical assumptions. In the revision, we will soften this to "highly decoupled" or "statistically independent". To quantitatively evaluate this separation, we define a Decoupling Score based on average absolute cross-correlation:
>
> $$Score = \frac{1}{K^2} \sum_{i=1}^K \sum_{j=1}^K |\text{Corr}(C_i, A_j)|$$
>
> Furthermore, to measure non-linear dependence between branches, we evaluated the Latent Mutual Information (MI):
>
> $$\text{Latent MI} = \frac{1}{K^2} \sum_{i=1}^K \sum_{j=1}^K I(C_i; A_j)$$
>
> where $I(\cdot;\cdot)$ denotes the Shannon mutual information between the $i$-th and $j$-th principal components of $C$ and $A$. The results remain consistently low across datasets (e.g., MSRSTrain: Vis 0.0503, IR 0.0601; Kaist: Vis 0.0981, IR 0.1340), confirming effective functional separation (see [Decoupling](https://anonymous.4open.science/r/Anonymous-C632/Reviewer_YjME/Decoupling.png)).
>
> # [Q2, W2]: Quantitative Manifold Metrics
>
> We agree that t-SNE alone is insufficient as it emphasizes local structure. Following your suggestion, we computed standard manifold learning metrics: Trustworthiness ($TW$) and Continuity ($CT$). They quantify the extent to which $k$-nearest neighbor rankings are preserved between the original input space and the latent representations:
>
> $$TW(k) = 1 - C \sum_{i=1}^N \sum_{j \in U_i^{(k)}} (r(i,j) - k)$$
>
> $$CT(k) = 1 - C \sum_{i=1}^N \sum_{j \in V_i^{(k)}} (\hat{r}(i,j) - k)$$
>
> |**Dataset**|**Vis TW**|**Vis CT**|**IR TW**|**IR CT**|
> |-|-|-|-|-|
> |Kaist|0.7651|0.7959|0.8367|0.8250|
> |M3FD|0.8065|0.7146|0.6822|0.6839|
> |MSRS|0.8237|0.8550|0.7441|0.7567|
> |TNO|0.7362|0.6736|0.5880|0.5617|
>
> These quantitative results explicitly confirm that our encoding process preserves intrinsic geometric relationships without severe spatial distortions, justifying our topology-related claims (see [Manifold](https://anonymous.4open.science/r/Anonymous-C632/Reviewer_YjME/Manifold.png)). We will include this formal analysis in the revision.
>
> # [Q3]: Adversarial Constraints Wording
>
> We agree with your insight. Adversarial training encourages distribution matching rather than guaranteeing strict manifold membership. We will adjust the wording accordingly to ensure technical precision.
>
> # [Q4]: Mechanism and Motivation of Attribute Interpolation
>
> We apologize if our initial description was ambiguous; our framework does not perform rigid linear interpolation. As defined in Sec 3.4, the stochastic factor $\alpha$ is injected as a prior bias into the unbounded logit domain. The final gating weight $\boldsymbol{\mu}$ dynamically shifts based on adaptive intrinsic channel priors, yielding smooth, highly non-linear transitions. Physically, traversing this attribute manifold simulates continuous environmental shifts. Methodologically, this stochastic traversal serves as a manifold regularization strategy: by forcing the decoder to reconstruct scenes under diverse and even extreme attribute states (e.g., via Boundary Bias sampling), we compel the content encoder to maintain robust, invariant semantic representations. This ensures that the structural geometry remains decoupled from transient environmental variations, preventing the model from collapsing into trivial identity mappings when encountering unseen modal characteristics. (See [Interpolation](https://anonymous.4open.science/r/Anonymous-C632/Reviewer_YjME/Interpolation.png))
>
> # [Q5]: Attribute Role Separation
>
> To conclusively prove $C$ and $A$ are functionally distinct, we evaluated replacing extracted attributes with zero tensors or shuffling them with randomly mismatched references [Tables](https://anonymous.4open.science/r/Anonymous-C632/Reviewer_YjME/Tables.png), [Visuals](https://anonymous.4open.science/r/Anonymous-C632/Reviewer_YjME/Visuals.png). Zeroing $A$ causes metrics to crash precipitously (e.g., MSRS $Q_{abf}$: $0.66 \to 0.22$, MI: $2.98 \to 1.69$), outputting flat structural outlines devoid of thermal or illumination saliency. This confirms $A$ exclusively controls modality appearance. Conversely, shuffling $A$ keeps spatial geometry visually stable, which proves that $C$ independently retains structure. However, this mismatch degrades semantic coherence and quantitative metrics (e.g., Kaist $Q_{abf}$: $0.70 \to 0.62$). Injecting mismatched attributes misaligns semantics but fundamentally preserves base geometry. We will include this in the revision.
>
> # [Discussion on Limitations]
>
> Fixed-size patch training inherently stabilizes optimization by filtering uninformative low-contrast regions. Please refer to our response to **Reviewer 2yxB limitations** for discussions on other potential limitations.
>
> -----
> Thank you once again for your time and effort. We would be glad to incorporate these insightful discussions into a revision.

---

> > ### Author Rebuttal · Reviewer_YjME · 2026-04-02
> >
> > I keep my positive view to this paper since the authors have solved my concerns.

---

> > > ### Author Response · Authors · 2026-04-06
> > >
> > > We appreciate your continued positive view of our paper. Thank you for confirming that the provided clarifications resolved your initial concerns. The helpful feedback you shared during the review process has offered a valuable perspective. We plan to integrate your suggestions into the manuscript, which will likely enhance the overall presentation of our work. Thank you for your time and thoughtful evaluation.

---

### Official Review · Reviewer_uDFv · 2026-03-12

**Soundness:** 3
**Presentation:** 3
**Significance:** 3
**Originality:** 2
**Overall Recommendation:** 4
**Confidence:** 4

**Summary:**

This paper proposes LaRA-Fusion, an unsupervised infrared-visible image fusion framework built around a dual-loop manifold constraint design. The method explicitly disentangles input representations into shared content anchors and modality-specific attributes, and then constrains them through an inner loop for geometric invertibility and an outer loop for adversarial manifold anchoring. In addition, the framework introduces a Manifold Rectification Block (MRB) for content aggregation, a Hierarchical Attribute Arbitration (HAA) mechanism for adaptive attribute fusion with stochastic interpolation, and a Modulated Synthesis Block (MSB) for attribute-guided reconstruction. The paper evaluates the method on four public IVIF benchmarks, including in-domain testing on MSRS and cross-dataset testing on TNO, Kaist, and M3FD, and reports consistent improvements over a range of recent baselines, together with ablations and some interpretability analysis.
This research aims to focus on a notable topic in IVIF, namely how to improve robustness and representation quality in unsupervised fusion where no ideal ground truth is available. A core issue considered by the paper is whether translation/reconstruction-based unsupervised objectives can induce degenerate shortcuts in the latent space, and whether explicit manifold-oriented constraints can alleviate this issue. This problem framing is relevant and potentially useful for the broader fusion community.

**Compliance With Llm Reviewing Policy:**

Affirmed.

**Final Justification:**

The authors have almost addressed my previous concerns, so I would keep my positive view to this paper as 4.

**Key Questions For Authors:**

1. The central narrative of the paper is that existing unsupervised IVIF methods suffer from “degenerate numerical shortcuts” and off-manifold drift. Can the authors provide more direct experimental evidence beyond the current reconstruction residuals and similarity analyses to support this diagnosis? For example, could they design a controlled experiment to explicitly show that competing methods indeed encode subtle source-modality-specific artifacts in the translated states?
2. The paper repeatedly uses topology- and manifold-related terminology. Under what precise definition does the claimed “topological consistency” hold? Is this intended mainly as an intuitive description, or do the authors mean it as a more rigorous formal claim? If the latter, please clarify the underlying assumptions and specify what properties are actually preserved.
3. Since part of the paper’s motivation is more stable semantic preservation, could the authors further evaluate the fused images on downstream detection or segmentation tasks, for example under MSRS or M3FD-related settings, to verify whether the method outperforms representative baselines? Such evaluation would significantly strengthen the paper’s argument for practical significance and impact.
4. Some components appear to have partially overlapping functionality, especially MRB, HAA, and MSB, all of which may affect representation quality and synthesis performance. Could the authors more clearly disentangle their respective roles, i.e., which module mainly contributes to geometric preservation, which to modality balancing, and which to artifact suppression? In addition, while it is appreciated that the paper compares against several recent baselines, could the authors clarify whether all baselines were reproduced or evaluated under fully consistent training and testing settings? A more detailed protocol description would help strengthen the fairness of the comparisons.

**Limitations:**

The paper still has several limitations. First, its strongest claims regarding topology preservation and shortcut elimination are only partially validated and remain more interpretive than rigorously demonstrated. Second, the experimental evaluation relies mainly on standard IVIF metrics and visual comparisons, which are not yet sufficient to fully support the broader robustness claims. Third, the method itself is fairly complex, involving multiple interacting modules and loss terms, so its training sensitivity, reproducibility, and computational cost warrant further examination. Fourth, although the cross-dataset results are encouraging, the paper has not yet demonstrated robustness under more explicit or severe distribution shifts. Finally, the originality of the work is solid but not fundamental, as it mainly builds on the integration of several existing ideas.

**Strengths And Weaknesses:**

Strengths：
1. The paper addresses a meaningful problem in unsupervised IVIF. It highlights that reconstruction- or translation-based unsupervised training can lead to shortcut solutions and weak semantic alignment in latent space. This concern is well motivated, and the discussion of “degenerate numerical shortcuts” and topological inconsistency gives the paper a clearer conceptual foundation than many purely engineering-driven fusion works.
2. Some methodological claims are overstated or insufficiently specified. Although the paper uses terms such as “manifold boundary,” “geodesic path,” and “topological inconsistency,” the actual method mainly relies on adversarial training, feature aggregation, interpolation regularization, and reconstruction losses. As a result, the theoretical framing appears stronger than the level of formal analysis actually provided.
3. The empirical results are competitive and generally strong on the reported metrics. The paper shows consistent improvements over baselines across four datasets, including cross-dataset evaluation without fine-tuning. This is encouraging, especially because cross-dataset performance is important for robustness in IVIF.
4. In IVIF, many methods improve standard fusion metrics while offering limited evidence that the learned representation is genuinely more stable. This work at least tries to go one step further by linking stochastic interpolation, manifold regularization, and latent-space behavior to generalization under environmental shifts. That direction is worthwhile.

Weaknesses：
1. Some methodological claims are not fully specified and appear somewhat overstated. Although the paper adopts terms such as “manifold boundary,” “geodesic path,” and “topological inconsistency,” the actual method is mainly built on adversarial training, feature aggregation, interpolation regularization, and reconstruction losses. As a result, the theoretical narrative is stronger than the formal analysis currently provided.
2. The paper uses “manifold” extensively, but in practice the manifold seems to function more as an implicit latent space than as an explicitly defined, modeled, or measurable object. Key questions remain unclear: how is this manifold represented, how are its boundaries defined, what constitutes in-manifold versus out-of-manifold samples, and how can one quantitatively verify that the method stays closer to the “correct” manifold? Without clearer answers, the claimed “manifold preservation” reads more like a conceptual motivation than a rigorously established technical property.
3. A common challenge in IVIF is that the relative importance of infrared and visible modalities varies across scenes: infrared may dominate target saliency, while visible images may contribute more to fine textures. Although the paper introduces HAA for attribute arbitration, it does not sufficiently analyze its behavior under highly imbalanced conditions, such as visible overexposure, low-contrast infrared, or extremely weak local thermal targets. Without such analysis, the functional boundary of the arbitration module remains unclear.
4. The full system combines MRB, HAA, MSB, the inner and outer loops, and multiple loss terms, which improves modeling capacity but also raises the concern that the gains may come more from system complexity than from a clearly dominant new idea. Given the number of interacting components, the paper should more clearly disentangle which modules are truly essential and whether comparable performance could be achieved with a simpler design.
5. The current results are obtained with a specific network architecture, but it remains unclear whether the dual-loop constraints would still be effective with a stronger or lighter backbone, and whether the gains from MRB, HAA, and MSB are stable across different architectural choices.

---

> ### Author Rebuttal · Authors · 2026-03-31
>
> We thank the reviewer for the positive assessment. Our responses follow:
>
> # [Q1, L1]: Evidence of Degenerate Numerical Shortcuts
>
> While extracting intermediate states from baselines is difficult, degenerate shortcuts leave observable spectral traces. We thus conducted 1D Power Spectral Density (PSD) analysis. Internally, our intermediate translations' PSD curves on MSRS closely approximate the natural $1/f$ decay without severe high-frequency anomalies (see [Internal PSD](https://anonymous.4open.science/r/Anonymous-C632/Reviewer_uDFv/Internal_PSD.png)). Externally, global average PSDs across test sets reveal several baselines exhibiting unnatural high-frequency uplifts (noise injection) or sags (texture collapse), whereas LaRA-Fusion maintains tighter alignment with the authentic target manifold (see [Comparative PSD](https://anonymous.4open.science/r/Anonymous-C632/Reviewer_uDFv/Comparative_PSD.png)).
>
> # [Q2, W1, W2, L1]: Definition of Topological Consistency and Manifold
>
> We clarify that "topological consistency" in our manuscript serves primarily as an empirical and intuitive descriptor grounded in manifold learning. It denotes that the intrinsic local neighborhood structures and geometric relationships of the input modalities are preserved in the latent space without severe spatial distortion.
>
> To translate this into rigorously measurable properties, we evaluated Trustworthiness and Continuity, which quantify how $k$-nearest neighbor rankings are preserved during encoding. For exact mathematical definitions, underlying assumptions, and comprehensive quantitative results confirming our dual-loop constraints effectively preserve original neighborhood structures, please refer to our response to **Reviewer YjME Q2**. We will include these definitions in the revision.
>
> # [Q4, W3, W4, L3, L4, L5]: Module Disentanglement, Imbalanced Conditions, and Fairness Protocol
>
> Each module has a strictly orthogonal purpose. The MRB mitigates cross-modal topological misalignment via residual feature aggregation to preserve spatial structure. The HAA dynamically arbitrates multi-scale attributes to adapt to varying scene conditions. The MSB employs attribute-guided weight modulation/demodulation, explicitly eliminating decoding artifacts associated with standard normalization. Thus, they transcend mere integration of existing ideas to resolve specific topological bottlenecks.
>
> Regarding highly imbalanced conditions, we explicitly analyzed this in Appendix C (Table 3). Even when processing low-contrast infrared (e.g., sample 00720N) or illumination-constrained visible inputs (e.g., 01244N), the HAA robustly arbitrates attributes. It maintains a strictly monotonic correlation between the attribute vector norm and the physical Shannon entropy of the scene, outputting structurally stable intrinsic priors without functional collapse.
>
> Regarding evaluation fairness, we utilized official pre-trained weights for all baselines without re-tuning. For standard IVIF metrics, we evaluated default model outputs directly. For downstream semantic tasks, we applied a consistent pseudo-color rendering protocol using chrominance channels from the source visible images to baselines natively outputting grayscale images. This exact protocol was uniformly applied to our method and all baselines to ensure rigorous fairness.
>
> # [Q3, L2]: Downstream Task Evaluation
>
> We conducted evaluations using YOLO on M3FD and DeepLabV3+ on MSRS. To mitigate severe MSRS long-tail bias (where rare classes yield zero), we used an adjusted mIoU. Under this fair setting, LaRA-Fusion achieves a competitive adjusted mIoU of **65.53** alongside the highest mAP@50 (**66.74**) in object detection. We refer you to our response to **Reviewer VYpc Q2** for complete quantitative tables and qualitative visualizations.
>
> # [W5, L3]: Architectural Coupling and Backbone Generalization
>
> Directly replacing the entire network with an off-the-shelf monolithic backbone (e.g., standard ViT) is not feasible due to our highly coupled, disentangled dual-stream paradigm. Specifically, inputs are separated into a shared spatial content anchor ($\textbf{C}$) and a global attribute vector ($\textbf{A}$). Our specialized modules are strictly tied to this orthogonal routing: MRB exclusively processes concatenated spatial content ($\textbf{C}_{cat}$), HAA arbitrates 1D attributes, and MSB systematically integrates them using $\textbf{A}$ to modulate convolutional weights applied to $\textbf{C}$. A monolithic backbone would fundamentally disrupt this explicit disentanglement.
>
> However, internal feature extraction blocks within our Content and Attribute Encoders are flexible. They can be replaced with stronger or lighter blocks to scale computational cost, provided macroscopic dual-stream topology is maintained. We will clarify this boundary in the revision.
>
> ------
>
> Thank you once again for your time and effort. We will incorporate these discussions into the revision.

---

> > ### Author Rebuttal · Reviewer_uDFv · 2026-04-01
> >
> > The authors have almost addressed my previous concerns, so I would keep my positive view to this paper.

---

> > > ### Author Response · Authors · 2026-04-06
> > >
> > > Thank you for your continued positive assessment of our work. We are glad that the response adequately clarified your previous concerns. Your insightful comments have been highly beneficial. We hope to carefully integrate these proposed improvements into the manuscript to better reflect your feedback. We sincerely appreciate the time and effort you have put into reviewing our paper.

---

### Official Review · Reviewer_2yxB · 2026-03-13

**Soundness:** 4
**Presentation:** 3
**Significance:** 3
**Originality:** 3
**Overall Recommendation:** 5
**Confidence:** 4

**Summary:**

LaRA-Fusion, an unsupervised deep learning framework for Infrared and Visible Image Fusion (IVIF), is presented in this paper. The tendency for encoders to learn "degenerate numerical shortcuts," where they memorize high-frequency noise instead of extracting domain-invariant semantic structures to satisfy loss functions, is a serious weakness in the translation-based IVIF techniques currently in use. In order to address this, the framework projects inputs into a disentangled latent space that consists of unique attribute coordinates and a shared content anchor. Next, Dual-Loop Manifold Constraints are applied: an outer loop employs adversarial discriminators to anchor intermediate translations to the natural data manifold, while an inner loop uses self-reconstruction to enforce geometric reversibility. The approach, which is backed by specialized architecture such as the Manifold Rectification Block (MRB) and Hierarchical Attribute Arbitration (HAA), is thoroughly tested on the MSRS, Kaist, M3FD, and TNO datasets, exhibiting state-of-the-art performance in both quantitative metrics and visual fidelity.

**Compliance With Llm Reviewing Policy:**

Affirmed.

**Final Justification:**

The authors' comprehensive rebuttal has successfully resolved my technical reservations regarding the model's computational overhead and its manifold stability under stochastic perturbations. This directly reinforces my initial assessment, and I therefore maintain my rating of 5.

**Key Questions For Authors:**

1) How does the computational overhead (e.g., FLOPs, parameter count, and inference time) of LaRA-Fusion compare to baseline methods like DeFusion or LRRNet?
2) How sensitive is the model's structural stability to the choice of the scaling factor $\gamma=1.5$ and the noise variance $\sigma=0.2$ during the stochastic manifold exploration tests?

**Limitations:**

No, the authors have not adequately discussed the technical limitations of their work. While they provide a brief "Impact Statement" noting the general goal of advancing Machine Learning and stating there are no specific societal consequences to highlight, they omit technical failure cases.

Constructive Suggestion: The authors should add a dedicated limitations paragraph in the discussion or conclusion section addressing potential edge cases (e.g., performance in extreme weather conditions like heavy rain where both IR and VIS fail), hardware constraints, or limitations of the patch-based training scheme.

**Strengths And Weaknesses:**

Strengths:
- Originality & Significance: The paper proposes a novel "Dual-Loop" constraint (coupling geometric invertibility with adversarial boundaries) to solve the critical issue of degenerate numerical shortcuts and latent space collapse in unsupervised IVIF. Additionally, the stochastic manifold traversal mechanism offers a fresh perspective for maintaining structural stability.
- Soundness (Experimental Rigor): The technical approach is exceptionally rigorous, featuring a comprehensive ablation study that validates the necessity of the inner loop, outer loop, MRB, and HAA modules.
- Soundness (Interpretability): A standout interpretability analysis successfully maps the attribute vector norm directly to the physical Shannon entropy of the scenes. This provides compelling evidence that the latent space captures physical reality rather than arbitrary noise.
- Significance (Generalization): The model demonstrates impressive generalization by training solely on the MSRS dataset and achieving state-of-the-art results on Kaist, TNO, and M3FD test sets without any fine-tuning.Presentation: The submission is clearly written and logically structured. Excellent visual diagrams, particularly the t-SNE visualizations in Figure 10, effectively illustrate the model's structural disentanglement.

Weaknesses:
- Soundness (Computational Analysis): The paper lacks a detailed discussion on computational complexity, such as FLOPs, parameter counts, and inference speeds against baselines. This is a crucial omission since IVIF is often deployed in real-time, resource-constrained environments like autonomous driving.
- Presentation (Mathematical Density): Some mathematical formulations—specifically within the Manifold Rectification Block (MRB) and Hierarchical Attribute Arbitration (HAA) sections—are quite dense. The manuscript would benefit from more intuitive explanations connecting equations (e.g., Eqs. 2-4) to their direct geometric effects.

---

> ### Author Rebuttal · Authors · 2026-03-31
>
> We thank the reviewer for the positive assessment and constructive feedback on our "Dual-Loop" constraint, interpretability, and generalization. Our responses follow:
>
> # [Q1,W1]: Computational Overhead Comparison
>
> We rigorously evaluate all methods' complexity and mean inference time using $256 \times 256$ random tensors on an NVIDIA A800 GPU, reporting the average of 10 independent trials (100 runs each) measured via high-precision CUDA events following a 50-iteration warm-up:
>
> | **Method**         | **Params (M) ↓** | **FLOPs (G) ↓** | **Time (s) ↓** |
> | ------------------ | ---------------- | --------------- | -------------- |
> | DeFusion           | 7.8745           | 38.6390         | 0.0917         |
> | BDLFusion          | 32.7238          | 80.9720         | 0.1110         |
> | LRRNet             | 0.0492           | 6.0445          | 0.0395         |
> | CAF                | 0.1483           | 19.4532         | 0.0034         |
> | MMDRFuse           | 0.0001           | 0.0142          | 0.0003         |
> | ITFuse             | 0.0825           | 11.3514         | 0.1085         |
> | FusionBooster      | 0.5977           | 29.6726         | 0.0491         |
> | LUTFuse            | 0.0078           | 1.0339          | 0.0095         |
> | T2EA               | 0.2754           | 104.7457        | 0.0763         |
> | LaRA-Fusion (Ours) | 4.0820           | 53.5504         | 0.0618         |
>
> As shown, compared to models with heavier parameter and computational burdens like DeFusion and BDLFusion, LaRA-Fusion achieves lower inference latency. We acknowledge that certain ultra-lightweight architectures exhibit superior speed; however, these methods typically utilize specific acceleration techniques. For instance, MMDRFuse utilizes knowledge distillation, and LUTFuse distills features into learnable Look-Up Tables. In this context, LaRA-Fusion represents a principled trade-off, incurring a moderate and acceptable computational cost to achieve superior topological robustness and advanced fusion quality.
>
> # [Q2]: Sensitivity of Stochastic Manifold Exploration
>
> We completely agree that a model's structural stability should stem from its intrinsic architecture rather than specific hyperparameter choices (e.g., $\gamma=1.5, \sigma=0.2$). To objectively verify this, we conducted a rigorous **Continuous Parameter Sweep** across all datasets. Over 20 incremental steps, we systematically escalated the perturbation intensity by simultaneously scaling the attribute magnitude ($\gamma \in [1.0, 2.0]$) and injecting Gaussian noise ($\sigma \in [0.05, 0.5]$), and recorded the Mean Squared Error (MSE) of the latent feature drift. (please see the [trajectory visualizations](https://anonymous.4open.science/r/Anonymous-C632/Reviewer_2yxB/Continuous_Parameter_Sweep.png))
>
> The results reveal that as the perturbation intensity scales up, the average latent drift exhibits a progressive and strictly bounded growth. Even on the highly responsive MSRS dataset, the maximum drift trajectory remains tightly constrained within an absolute bound, without triggering feature divergence or geometric collapse. This confirms that our disentangled content anchors ($\textbf{C}$) are robust to a wide, continuous range of manifold shifts, indicating that the model is highly tolerant to diverse stochastic sampling parameters.
>
> # [W2, Limitations]: Mathematical Density and Technical Limitations
>
> To clarify W2: Geometrically, Eq.2 projects features to a joint manifold for cross-modal interaction. Eqs.3-4 then inject high-frequency residuals to repair topological gaps and preserve spatial topology. We will expand these links in the revision.
>
> Following your suggestion, we would explicitly add a dedicated "Limitations" paragraph to address the following aspects:
>
> - First, concerning extreme weather, while our evaluations under rainy conditions yield competitive visual results (please see the [rainy scenario](https://anonymous.4open.science/r/Anonymous-C632/Reviewer_2yxB/Rainy_Scenario.png)), we agree that in unexplored, catastrophic environments where both sensors simultaneously fail, our unsupervised constraints might inevitably struggle to extract valid semantics.
> - Second, regarding patch-based training, although utilized to actively filter out low-contrast uninformative regions, we acknowledge it inherently restricts capturing broader, image-level global contexts. To overcome this and further enhance the model's global representation capabilities, we plan to explore full-image training paradigms in future work.
> - Finally, regarding hardware constraints, our model contains only 4.08M parameters. This relatively compact parameter space ensures a highly manageable memory footprint during inference on standard GPUs, effectively balancing the moderate computational FLOPs.
>
> ------
>
> Thank you once again for your time and effort. We would be glad to incorporate these insightful discussions into a revision.

---

> > ### Author Rebuttal · Reviewer_2yxB · 2026-04-04
> >
> > The authors have addressed my technical questions regarding computational overhead and manifold stability with high-quality data. While the rebuttal has improved the clarity and transparency of the work, it confirms my initial assessment of the paper's strengths rather than shifting its overall impact level. Therefore, I maintain my original recommendation.

---

> > > ### Author Response · Authors · 2026-04-06
> > >
> > > We appreciate your detailed feedback and the time you took to read our rebuttal. We are glad that the explanations regarding computational overhead and manifold stability addressed your technical questions. The points you raised highlighted specific areas where the clarity and transparency of our paper could be improved. We plan to incorporate these additional clarifications into the manuscript. Thank you again for your engagement throughout this process.

---

### Decision · Program_Chairs · 2026-04-30

**Decision:**

Accept (regular)

**Comment:**

All four reviewers converged on a positive assessment. Strengths consistently noted include the well-motivated problem framing around degenerate latent shortcuts, strong cross-dataset generalization, comprehensive ablations, and solid empirical results. The rebuttal effectively addressed the main concerns: computational overhead was quantified, manifold claims were backed by Trustworthiness/Continuity metrics, downstream detection and segmentation results were provided, and overstated claims (e.g., “orthogonal” subspaces) were appropriately softened.
Remaining limitations are minor: the theoretical grounding of manifold claims is empirical rather than formal, architectural complexity makes it difficult to identify a single dominant contribution, and the method’s behavior under severe distribution shifts is not fully characterized.
Overall, the paper makes a solid and useful contribution to unsupervised IVIF.